# Local field potentials reflect cortical population dynamics in a region-specific and frequency-dependent manner

Cecilia Gallego-Carracedo[1,2], Matthew G Perich[3,4†], Raeed H Chowdhury[5], Lee E Miller[6,7,8], Juan Álvaro Gallego[1]*

[1]Department of Bioengineering, Imperial College London, London, United Kingdom; [2]Neural and Cognitive Engineering Group, Centre for Automation and Robotics, Spanish National Research Council, Arganda del Rey, Spain; [3]Department of Neuroscience, Icahn School of Medicine at Mount Sinai, New York, United States; [4]Département de neurosciences, Faculté de médecine, Université de Montréal, Montréal, Canada; [5]Department of Bioengineering, University of Pittsburgh, Pittsburgh, United States; [6]Department of Neuroscience, Northwestern University, Chicago, United States; [7]Department of Biomedical Engineering, Northwestern University, Evanston, United States; [8]Department of Physical Medicine and Rehabilitation, Northwestern University, and Shirley Ryan AbilityLab, Chicago, United States

*For correspondence:
jgallego@imperial.ac.uk

Present address: †Département de neurosciences, Faculté de médecine, Université de Montréal, Montréal, Québec, Canada

**Abstract** The spiking activity of populations of cortical neurons is well described by the dynamics of a small number of population-wide covariance patterns, whose activation we refer to as 'latent dynamics'. These latent dynamics are largely driven by the same correlated synaptic currents across the circuit that determine the generation of local field potentials (LFPs). Yet, the relationship between latent dynamics and LFPs remains largely unexplored. Here, we characterised this relationship for three different regions of primate sensorimotor cortex during reaching. The correlation between latent dynamics and LFPs was frequency-dependent and varied across regions. However, for any given region, this relationship remained stable throughout the behaviour: in each of primary motor and premotor cortices, the LFP-latent dynamics correlation profile was remarkably similar between movement planning and execution. These robust associations between LFPs and neural population latent dynamics help bridge the wealth of studies reporting neural correlates of behaviour using either type of recordings.

## Editor's evaluation

This paper will be of interest to electrophysiologists, systems neuroscientists and neural engineers. The authors describe a framework for evaluating the comparison between LFP dynamics and spikes and perform this comparison for several datasets recorded from motor, premotor, and sensory areas of cortex in rhesus macaque monkeys. These results serve as an important benchmark for the information content of LFP recordings, which is relevant to data collection in neuroscientific investigations and to designing brain computer interfaces.

**Figure 1.** Hypothesis. (**A**) Microelectrode arrays record both the spiking activity of single neurons (denoted as N1, N2, N3) and the local field potentials at these same sites (denoted as LFP1, LFP2, LFP3) of a cortical region X. (**B**) Synchronisation of synaptic currents within this circuit generate LFPs in different bands (top). Circuit connectivity and other biophysical properties also constrain the coordinated latent dynamics of the N1, N2, N3 population (bottom). We thus hypothesise that, for each brain region, there will be a frequency-dependent association between LFPs and latent dynamics that should remain stable while circuit biophysics remain stable.

## Introduction

Researchers and clinicians often monitor brain activity using implanted electrodes to understand how the brain drives behaviour and to improve clinical outcomes (**Willett et al., 2021**; **Flesher et al., 2021**; **Quian Quiroga, 2019**; **Rutishauser et al., 2021**). Modern multielectrode arrays allow the spiking activity of hundreds of neurons to be observed simultaneously (**Hong and Lieber, 2019**). These same electrodes can also capture the lower frequency local field potentials (LFPs) that result from synaptic currents summed across many thousands of neurons (**Mitzdorf, 1985**; **Einevoll et al., 2013**; **Lindén et al., 2011**; **Buzsáki et al., 2012**; **Pesaran et al., 2018**). Uncovering a relationship between the collective dynamics of populations of single neurons and those of larger scale LFPs will help bridge studies looking at each of these signals in isolation and advance our understanding of the human brain.

The spiking activity of neural populations can be well characterised by dynamics of relatively few covariance patterns (**Vyas et al., 2020**; **Gallego et al., 2017**), which we refer to as the 'latent dynamics' (**Pandarinath et al., 2018**; **Gallego et al., 2020**). Studying these latent dynamics has provided insight into aspects of animal behaviour that was not apparent from the activity of single neurons alone, including decision making (**Machens et al., 2010**; **Mante et al., 2013**), movement planning (**Churchland et al., 2010**; **Kaufman et al., 2014**; **Dekleva et al., 2018**), learning (**Sadtler et al., 2014**; **Oby et al., 2019**; **Perich et al., 2018**; **Vyas et al., 2018**; **Sun et al., 2020**), the control timing (**Dekleva et al., 2018**; **Wang et al., 2018b**; **Remington et al., 2018**), and production of consistent behaviour (**Gallego et al., 2020**).

The latent dynamics are constrained by the neural covariance structure, which is shaped in part by circuit connectivity (**Sadtler et al., 2014**; **Oby et al., 2019**; **Okun et al., 2015**; **Feulner and Clopath, 2021**; **Feulner et al., 2021**). Currents through these same connections as well as other biophysical properties of the circuit are the main contributors to the generation of the LFPs (**Mitzdorf, 1985**; **Einevoll et al., 2013**; **Lindén et al., 2011**; **Buzsáki et al., 2012**; **Pesaran et al., 2018**; **Figure 1A**). In particular, correlations in the synaptic input currents across the population yield changes in LFP power at specific frequency bands, which often relate to specific brain functions (**Buschman et al., 2012**; **Tremblay et al., 2015**; **Liu and Newsome, 2006**; **Pesaran et al., 2002**; **Scherberger et al., 2005**; **Gail et al., 2004**; **Holmes et al., 2018**; **Sanes and Donoghue, 1993**; **Zhuang et al., 2010**; **Bansal et al., 2012**; **Flint et al., 2012**; **Stavisky et al., 2015**; **Perel et al., 2015**; **Baker et al., 1997**; **Witham et al., 2010**; **Witham and Baker, 2012**; **Buzsáki, 2002**). For example, in the motor cortices,

movement-related information lies primarily at low and high frequencies (*Zhuang et al., 2010*; *Bansal et al., 2012*; *Flint et al., 2012*; *Stavisky et al., 2015*) (i.e., <5 Hz and >50 Hz), yet, the clearest LFP signature of imminent movement initiation is a marked decrease in activity in the 15–30 Hz band (*Sanes and Donoghue, 1993*; *Perel et al., 2015*; *Donoghue et al., 1998*; *Witham et al., 2007*). In somatosensory cortical areas 2 and 3a, this same 15–30 Hz band contains signatures potentially related to afferent input from the limb (*Witham et al., 2010*). It seems likely that these region-specific LFP phenomena relate to the neural population latent dynamics of each region, as they are both ultimately driven by the same synaptic currents.

Previous studies investigated the relationship between single neuron activity and LFPs (*Pesaran et al., 2002*; *Perel et al., 2015*; *Witham and Baker, 2012*; *Ray et al., 2008*; *Ray and Maunsell, 2011*; *Rule et al., 2017*; *Murthy and Fetz, 1996b*). However, we lack a systematic description of how the different LFP bands relate to the latent dynamics reflecting the coordinated activity of the neural populations driving behaviour. Here, we report on the region-specific relationship between the LFPs and the latent dynamics by addressing four hypotheses. First, since both LFP (*Mitzdorf, 1985*; *Einevoll et al., 2013*; *Lindén et al., 2011*; *Buzsáki et al., 2012*; *Pesaran et al., 2018*) and latent dynamics (*Sadtler et al., 2014*; *Oby et al., 2019*; *Okun et al., 2015*) are likely shaped by the circuit biophysics (*Figure 1B*), we expect to find a robust relationship between these two signals. Second, we anticipate finding fundamental differences across LFP bands since only a specific subset of them is correlated with behaviour for any given brain region (*Buschman et al., 2012*; *Tremblay et al., 2015*; *Liu and Newsome, 2006*; *Pesaran et al., 2002*; *Scherberger et al., 2005*; *Gail et al., 2004*; *Holmes et al., 2018*; *Sanes and Donoghue, 1993*; *Zhuang et al., 2010*; *Bansal et al., 2012*; *Flint et al., 2012*; *Stavisky et al., 2015*; *Perel et al., 2015*; *Baker et al., 1997*; *Witham et al., 2010*; *Witham and Baker, 2012*; *Buzsáki, 2002*; *Murthy and Fetz, 1996a*). That is, the relationship between LFPs and latent dynamics should be frequency-dependent. Third, we expect that as long as circuit biophysics remain unchanged, the LFP-latent dynamics associations will remain similarly stable. Thus, the various associations between motor cortical LFPs and latent dynamics should not change in the short time scale spanning the preparation and execution of a movement. Lastly, since different sensorimotor cortical areas have important differences in inputs and function (*Kakei et al., 2001*; *Scott, 2012*; *Cisek and Kalaska, 2010*; *Das and Fiete, 2020*), as well as cytoarchitecture (*Hutsler et al., 2005*; *Dum and Strick, 1991*), we hypothesise that the relationships between LFPs and latent dynamics should be region-specific.

We tested these four hypotheses using intracortical recordings from three different regions of the primate sensorimotor cortex during the same reaching behaviour: (1) primary motor cortex (M1), the main cortical output that controls movement execution (*Rathelot and Strick, 2006*; *Fetz and Cheney, 1980*; *Morrow and Miller, 2003*; *Sergio et al., 2005*); (2) dorsal premotor cortex (PMd), a region that integrates inputs from many structures and is largely involved in movement planning (*Dekleva et al., 2018*; *Shen and Alexander, 1997a*; *Santhanam et al., 2009*; *Crammond and Kalaska, 1994*); and (3) area 2 of somatosensory cortex, where multimodal proprioceptive and cutaneous information about the state of the limb is processed (*Chowdhury et al., 2020*; *London and Miller, 2013*; *Pons et al., 1985*). Our results show that the relationship between the LFPs and the latent dynamics from the same cortical region is indeed frequency-dependent and varies across cortical regions. Yet, the region-specific LFP-latent dynamics 'correlation profiles' remain remarkably stable across the evolving processes mediating movement planning and execution (*Dekleva et al., 2018*; *Shen and Alexander, 1997b*; *Ames et al., 2019*). Finally, we show that these relationships do not trivially arise from correlations between single unit firing rates and LFP recordings from the same intracortical electrode; instead, they reflect the coordinated activity of neural populations. This holds both during movement execution and as animals perform more 'abstract computations' related to movement planning.

These results, which are critical for bridging studies of the sensorimotor system that use LFPs and neural population activity, paint a picture in which LFP bands relate to the latent dynamics in a stable, region-specific, and frequency-dependent manner. This picture has one unexpected feature: some LFP bands are strongly correlated with latent dynamics yet carry little information about the animal's movement. Given that low-dimensional latent dynamics have been found across many brain regions (*Gallego et al., 2017*; *Keemink and Machens, 2019*), our findings are likely to translate beyond the sensorimotor system. Such translation may be especially insightful for regions commonly studied using LFP rhythms, such as the hippocampus (*Buzsáki and Moser, 2019*).

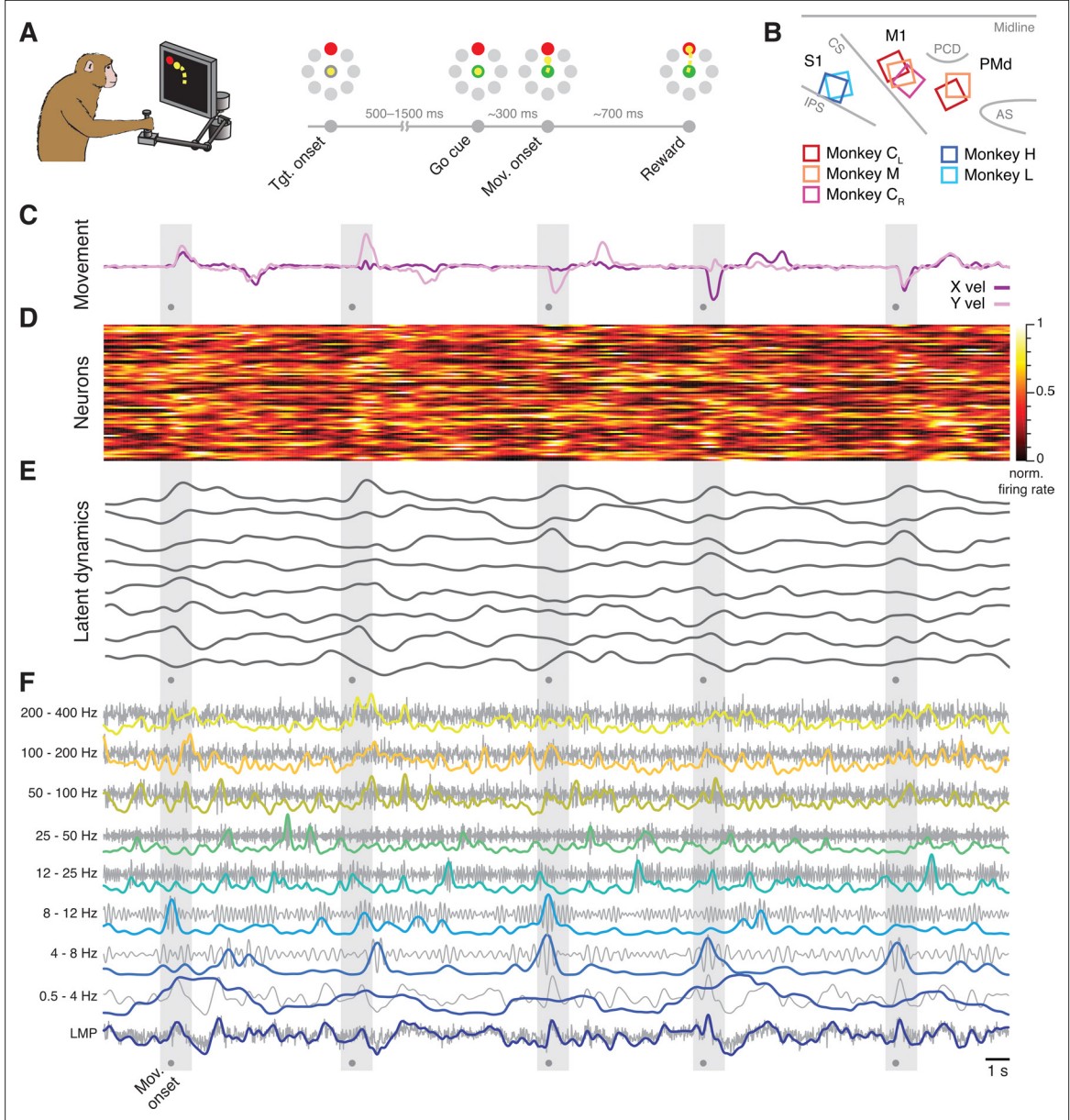

**Figure 2.** Behavioural task and neural recordings. (**A**) Monkeys performed a reaching task using a planar manipulandum. (**B**) Approximate locations of all seven arrays; each colour is one set of implants (legend). IPS, intraparietal sulcus; CS, central sulcus; AS, arcuate sulcus; PCD, precentral dimple. (**C**) Hand velocity along the horizontal (X) and vertical (Y) axes during five trials from a representative session from Monkey $C_L$. Shaded grey areas, reach epoch for each trial; dots, movement onset. (**D**) Example neural recordings showing the activity of 84 simultaneously recorded putative single neurons. (**E**) Latent dynamics corresponding to the neural firing rates in D. (**F**) Example local field potential (LFP) recordings. Each row shows the LFP activity filtered in one of the nine frequency bands we studied (grey) along with its power (colour). **A, B** have been adapted from Figure 2A and Figure 3A from *Gallego et al., 2020*.

## Results
### Behavioural task and neural recordings

We trained four monkeys (C, M, H, and L) to perform an instructed delay centre-out reaching task using a manipulandum (*Figure 2A*) (Methods). The monkeys started each trial by holding a cursor in the central target before one of the eight outer targets was presented. After a variable delay period, an auditory 'go cue' instructed them to move the cursor towards the presented target. A liquid reward was given after the monkeys had held the cursor in the target for a given period of time. For the monkeys with implants in area 2 (Monkeys H and L), both the delay and holding periods were omitted.

Each monkey was implanted with one or two 96-channel microelectrode arrays (*Figure 2B*). Monkeys C and M had dual implants in the arm regions of M1 and PMd, whereas Monkeys H and L had a single implant in the arm region of area 2. We also analysed a second dataset from the other (right) hemisphere of Monkey C, which we recorded in a different set of experiments; we denote this dataset as Monkey $C_R$, to distinguish it from the previous dual-region dataset from the same animal, which we denote as Monkey $C_L$.

We simultaneously recorded neural spiking and LFP on each electrode. The spiking signals were manually sorted to identify putative single neurons (*Figure 2D*). To compute the neural population latent dynamics, we constructed a high-dimensional neural state space in which the smoothed firing rate of each neuron was represented on a different axis; this way, the state of a population of *N* neurons at a given time *t* corresponds to one point in an *N*-dimensional state space. We used principal component analysis (PCA) to find the lower-dimensional neural manifold spanning the dominant population-wide activity patterns (*Gallego et al., 2017*; *Cunningham and Yu, 2019*) choosing the following manifold dimensionalities based on previous studies (*Gallego et al., 2020*; *Churchland et al., 2010*; *Perich et al., 2018*): 10 dimensions for M1, 15 for PMd, and 8 dimensions for area 2. We computed the latent dynamics by projecting the single neuron firing rates into each of the axes (principal components) that defined a given neural manifold (*Figure 2E*).

We calculated the LFP power in eight standard frequency bands (*Bansal et al., 2012*; *Flint et al., 2012*; *Stavisky et al., 2015*; *Zhuang et al., 2010*): 0.5–4 Hz, 4–8 Hz, 8–12 Hz, 12–25 Hz, 25–50 Hz, 50–100 Hz, 100–200 Hz, and 200–400 Hz for each electrode from which we could record at least one neuron. We also computed the local motor potential (LMP), which captures fluctuations in total LFP power (*Flint et al., 2012*; *Stavisky et al., 2015*; *Figure 2F*). For full details, see Methods section.

## The LFP-latent dynamics correlation profiles in primary motor cortex are frequency-dependent

We first investigated the similarity between the latent dynamics in M1 and each of the LFP frequency bands defined above. We used canonical correlation analysis (CCA), a method that quantifies the similarity between two sets of signals by finding the linear transformations that maximise their correlation (*Gallego et al., 2020*; *Bach and Jordan, 2002*; *Sussillo et al., 2015*) (Methods); we refer to this process simply as 'alignment'. For each session, we aligned the latent dynamics with the LFP power in each frequency band (*Figure 3A*). We performed this process separately for each recorded LFP signal, which yielded nine distributions of canonical correlation coefficients (CCs), one per frequency band, with as many samples as electrodes containing identified neurons (shown as individual data points in *Figure 3B*).

*Figure 3B* shows one representative session for each M1 monkey. We found that the relationship between LFPs and latent dynamics was both consistent across subjects and frequency-dependent: there were clear correlations between LFP and latent dynamics in the low and high bands (median correlation: 0.3–0.6), whereas the correlations approached zero in the mid-range bands (8–50 Hz) (coloured distributions in *Figure 3B*; *Figure 3—figure supplement 1* summarises all datasets). Importantly, these frequency-dependent LFP-latent dynamics correlations were also stable across sessions (*Figure 3—figure supplement 1*).

We devised a control analysis to verify that the larger correlations of the low and high frequency bands captured a significant relationship between LFPs and latent dynamics. Using tensor maximum entropy (TME) (*Elsayed and Cunningham, 2017*), we generated surrogate neural firing rates that both lay on the same neural manifold – that is, preserved the covariance across neurons – and had temporal statistics similar to the actual data (examples in *Figure 3—figure supplement 2*) (Methods). We reasoned that despite their spectral similarity, the correlations between the LFPs and the surrogate latent dynamics should be much lower than those between the LFPs and the actual latent dynamics. As predicted, the surrogate correlations were significantly lower than the actual correlations (median ≤0.1; p<0.001 for all comparisons, two-sided Wilcoxon's rank sum test; compare the black and coloured distributions in *Figure 3B*). Thus, the difference in LFP-latent dynamics correlations across LFP frequencies could not be trivially explained by similar spectral characteristics between the signals. We obtained similar LFP-latent dynamics correlation profiles for different manifold dimensionalities (*Figure 3—figure supplement 3*), and when pooling all putative single neurons on an electrode into 'multi-units' (*Figure 3—figure supplement 4A*). Our results also held when modifying the

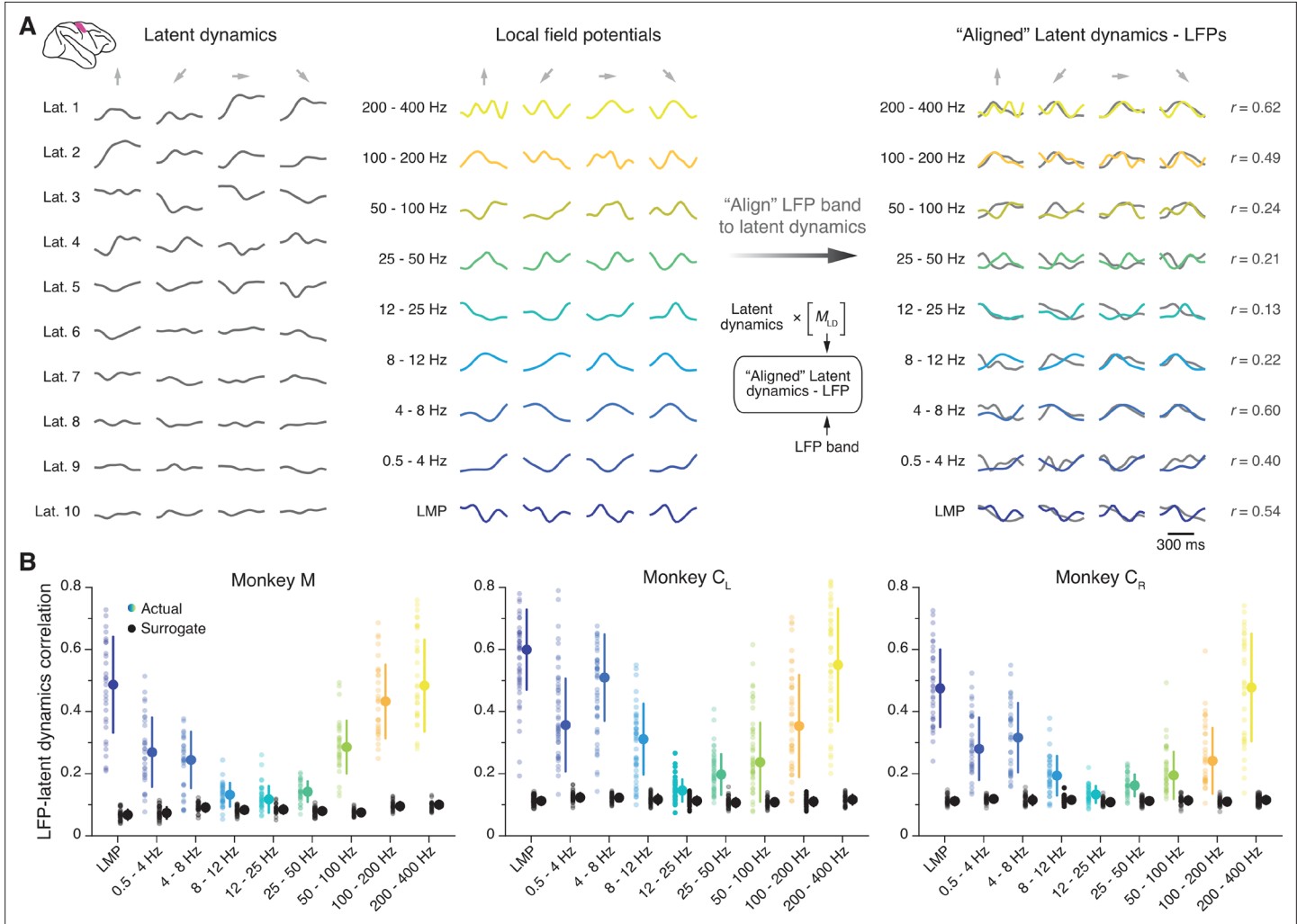

**Figure 3.** Similarity between primary motor cortex (M1) latent dynamics and each local field potential (LFP) band during movement execution. (**A**) Left: Example M1 latent dynamics during four reaches to different targets (direction indicated by the arrows above each column); figure shows top 10 dimensions of the latent dynamics. Middle: Example LFP power in each of the nine bands we study during the same three trials. Right: Canonical correlation analysis (CCA) 'alignment' finds strong similarities between certain LFP bands and the latent dynamics. r, correlation coefficient. From one session from Monkey $C_L$. (**B**) Correlation between each LFP band and the latent dynamics (coloured markers) during one representative session from each M1 monkey. Black markers show the control correlation values obtained after generating surrogate neural activity using tensor maximum entropy (TME); note that at low and high LFP frequencies, the actual correlations are much larger than the surrogate correlations. Error bars, median ± s.d. (n=31, 50, 34 for Monkeys M, $C_L$, and $C_R$, respectively).

The online version of this article includes the following figure supplement(s) for figure 3:

**Figure supplement 1.** Similarity between primary motor cortex (M1) latent dynamics and each local field potential (LFP) band.

**Figure supplement 2.** Surrogate latent dynamics control.

**Figure supplement 3.** The local field potential (LFP)-latent dynamics correlation profiles are preserved when changing neural manifold dimensionality.

**Figure supplement 4.** Further control analyses to illustrate the frequency-dependent relationship between primary motor cortex (M1) latent dynamics and each local field potential (LFP) band.

**Figure supplement 5.** The relationship between local field potential (LFP) bands does not predict their association with the latent dynamics.

**Figure supplement 6.** The latent local field potential (LFP) signals have a similar association with the neural population latent dynamics as the individual LFP signals do.

specific frequency bands used for LFP pre-processing within reasonable intervals (*Figure 3—figure supplement 4B*), and could not be explained by differences in LFP variance across bands (*Figure 3— figure supplement 4C*), or by differences in the LFP-LFP relationship across bands (*Figure 3—figure supplement 5*). To confirm our results were not biased by the dimensionality reduction applied to the

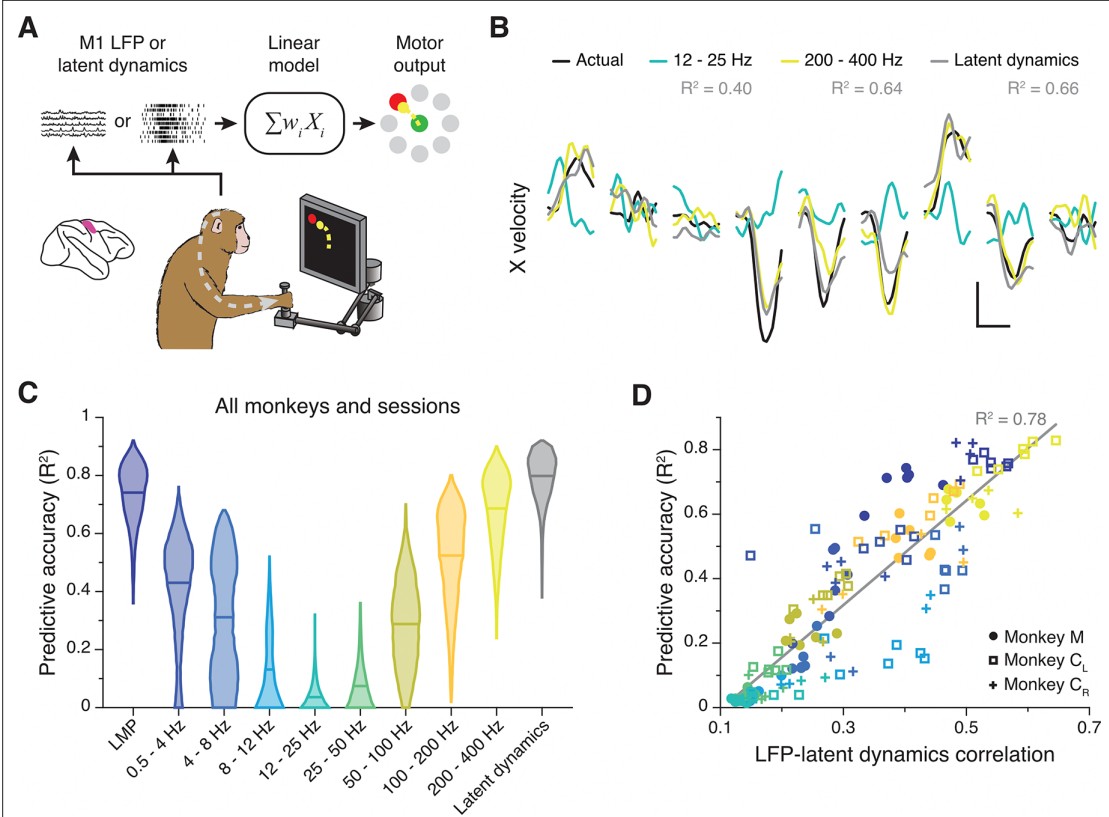

**Figure 4.** Decoding movement kinematics from primary motor cortex (M1) local field potentials (LFPs). (**A**) We trained linear decoders to predict hand velocity in the x and y axes from the either the latent dynamics or the LFP power in different frequency bands. (**B**) Example x-axis velocity predictions during nine randomly selected trials from one session from Monkey $C_L$. Note the clear difference in predictive accuracy between the example LFP bands (values for each input signal during the example trials shown are indicated in the legend). Scale bars: horizontal, 300 ms; vertical, 10 cm·s$^{-1}$. (**C**) Predictive accuracy of each LFP band (colours) and the latent dynamics (grey) pooled across all M1 monkeys and sessions. Violin, probability density for one frequency band (n=1600); horizontal bars, median. (**D**) Linear relationship between the M1 LFP-latent dynamics correlation coefficients and LFP predictive accuracy. Each marker denotes one LFP band from one session; different monkeys are shown using different markers (legend). Markers are colour-coded as in C. Grey line, linear fit to pooled data and goodness of fit (R$^2$).

The online version of this article includes the following figure supplement(s) for figure 4:

**Figure supplement 1.** Additional analyses to illustrate movement decoding from primary motor cortex (M1) local field potential (LFP).

single neuron firing rates, we computed the 'latent LFP signals' that explained most of the variance of the multi-channel LFP recordings within each frequency band (Methods). These latent LFP signals had the same frequency-dependent relationship with the neural population latent dynamics as the full-dimensional LFP power in each channel (*Figure 3—figure supplement 6*).

M1 latent dynamics (*Pandarinath et al., 2018*; *Gallego et al., 2020*) and some LFP bands (*Zhuang et al., 2010*; *Bansal et al., 2012*; *Flint et al., 2012*; *Stavisky et al., 2015*) allow accurate prediction or 'decoding' of movement kinematics. We thus asked whether the strength of the LFP-latent dynamics correlation anticipates how well an LFP band predicts behaviour. We used standard linear decoders to predict hand velocity either from all electrodes within each LFP band or from the latent dynamics (*Gallego et al., 2020*; *Flint et al., 2012*; *Figure 4A*) (Methods). Decoding accuracy varied considerably across LFP bands, with some (e.g., the LMP and 200–400 Hz) being nearly as predictive as the latent dynamics (*Figure 4B and C*; *Figure 4—figure supplement 1A*). The accuracy of each LFP band was strongly correlated with its similarity to the latent dynamics (R$^2$=0.78; p<0.001, F-test; see *Figure 4D*), and all of the accurate decoders made similar predictions (pairwise correlation: >0.6; *Figure 4—figure supplement 1B*). However, the predictive accuracy of an LFP band did not depend on its variance, since it was similar across all bands (*Figure 3—figure supplement 4C*). Predicting hand velocity from individual LFP channels was notably worse than predicting from all the channels (compare *Figure 4C* and *Figure 4—figure supplement 1C*; also see *Figure 4—figure supplement*

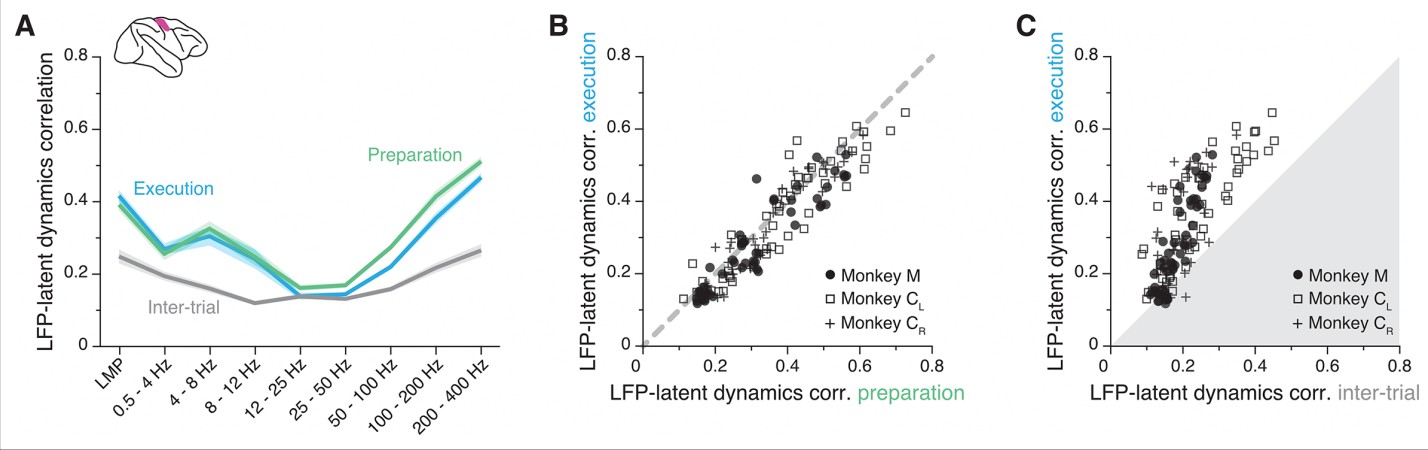

**Figure 5.** The primary motor cortex (M1) local field potential (LFP)-latent dynamics correlation profile is preserved between movement planning and execution. (**A**) LFP-latent dynamics correlation profiles across all sessions from all three M1 monkeys. Line and shaded areas, mean ± s.e.m. across all session medians (n=16). Each epoch is shown in a different colour. (**B**) Comparison between the LFP-latent dynamics correlations during movement preparation and execution. Each marker shows one frequency band for one session; each monkey is represented using a different marker (legend). Note the very strong similarity between epochs. (**C**) Comparison between the LFP-latent dynamics correlations during movement execution and the inter-trial period. Data formatted as in B. Note the marked decrease in correlations during the inter-trial period.

The online version of this article includes the following figure supplement(s) for figure 5:

**Figure supplement 1.** Decrease in primary motor cortex (M1) firing rates and neural correlations when monkeys are not engaged in the task.

*1D*). Thus, for M1, the relationship between LFP power and latent dynamics varies across frequency bands and predicts how strongly each band relates to the ongoing motor output.

## The LFP-latent dynamics correlation profiles are stable between movement planning and execution

We identified a robust frequency-dependent relationship between M1 LFPs and latent dynamics during movement execution. Since M1 is also involved in movement planning (*Dekleva et al., 2018*; *Shen and Alexander, 1997a*; *Riehle and Requin, 1989*), we asked whether this relationship remains stable between these two processes underlying behaviour. Repeating the previous CCA alignment procedure for the instructed delay epoch showed that the LFP-latent dynamics correlation profile was virtually identical between movement planning and execution (*Figure 5A*): a correlation analysis between all pairs of LFP-latent dynamics correlations across all frequency bands, sessions, and monkeys revealed a strong linear association ($R^2$=0.87; p<0.001, F-test; *Figure 5B*).

To provide a reference for the observed correlations, we repeated the alignment procedure during the inter-trial interval, when monkeys were not actively planning or executing the behaviour. In this period, M1 is not actively engaged in movement generation so neurons typically fire less (*Figure 5—figure supplement 1A,B*) and become less correlated (*Figure 5—figure supplement 1C*). This is due in large part to the lack of strong synaptic currents driving the population. Since synaptic currents are the main contributors to LFP generation (*Buzsáki et al., 2012*), we predicted that the LFP-latent dynamics correlations should decrease during the inter-trial interval when M1 is less actively engaged in producing behaviour. As expected, the correlations indeed became much lower during the inter-trial intervals than during movement execution (*Figure 5A and C*).

The observation that the LFP-latent dynamics correlation profile remains preserved between movement planning and execution extends our previous findings to a more abstract process occurring in the absence of movement. It also indicates that the observed association is not a trivial epiphenomenal consequence of the frequency content of movement, but instead likely reflects underlying physiological processes related to the production of behaviourally relevant neural activity. The decrease in LFP-latent dynamics correlations during the inter-trial intervals provides additional support for this interpretation. Even though the exact circuit architecture is likely stable between these periods, M1 is not actively engaged in generating behaviour. As expected, the absence of synaptic inputs and the

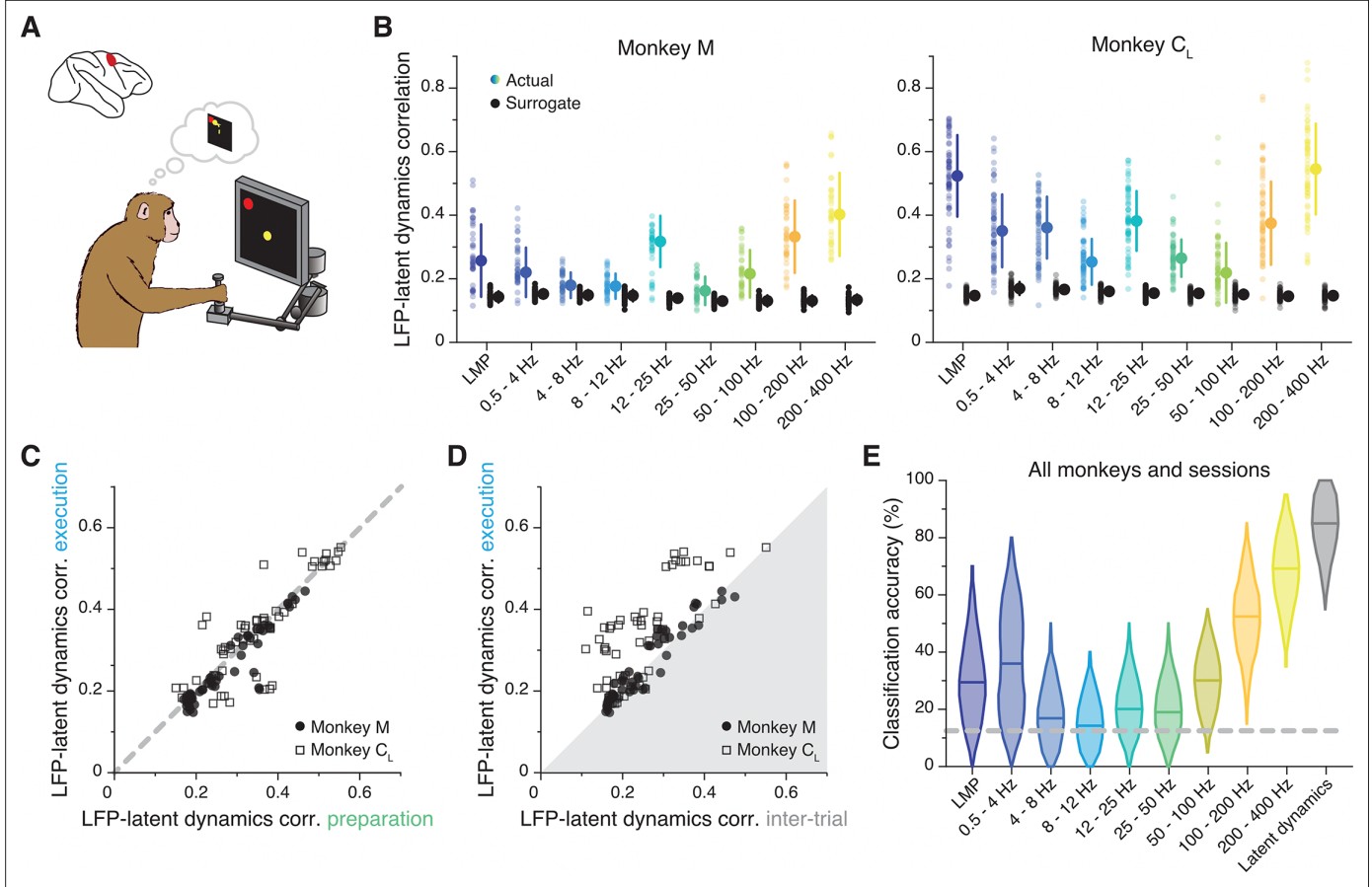

**Figure 6.** Similarity between dorsal premotor cortex (PMd) latent dynamics and each local field potential (LFP) band during movement preparation. (**A**) We focus on the instructed delay period following target presentation and preceding the subsequent go cue. (**B**) Correlation between each LFP band and the latent dynamics (coloured markers) during one representative session from each PMd monkey. Black markers show the control correlation values obtained after generating surrogate neural activity using tensor maximum entropy (TME). Error bars, median ± s.d. (n=40, 63 for Monkeys M and $C_L$, respectively). (**C**) Comparison between the LFP-latent dynamics correlations during movement preparation and execution. Each marker shows one frequency band for one session; each monkey is represented using a different marker (legend). Note again the very strong similarity between epochs. (**D**) Comparison between the LFP-latent dynamics correlations during movement execution and the inter-trial period. Data formatted as in C. (**E**) Accuracy of classifiers that predict the reach direction based on each LFP band (colours) and the latent dynamics (grey); data pooled across all PMd monkeys and sessions. Violin, probability density for one frequency band (n=600); horizontal bars, median; dashed horizontal line, chance level.

The online version of this article includes the following figure supplement(s) for figure 6:

**Figure supplement 1.** Similarity between dorsal premotor cortex (PMd) latent dynamics and each local field potential (LFP) band, and predicting reach direction from PMd LFPs.

**Figure supplement 2.** Similarity of local field potential (LFP) spectral properties across frequency bands and behavioural epochs.

resultant reduction in LFP and neural population activity during these intervals in which monkeys were not engaged in the task greatly reduced the LFP-latent dynamics correlations in M1.

## The LFP-latent dynamics correlation profiles change between primary motor and premotor cortices

The previous sections demonstrated a frequency-dependent relationship between the LFPs and latent dynamics in M1. We next asked whether similar associations could be found in other sensorimotor cortical regions. We studied PMd, a 'higher' motor region with a different cytoarchitecture from M1 (*Dum and Strick, 1991*) thought to be key for planning an action (*Dekleva et al., 2018*; *Crammond and Kalaska, 1994*; *Shen and Alexander, 1997b*; *Ohbayashi et al., 2016*; *Churchland and Shenoy, 2007*). We repeated the comparison between latent dynamics and LFP bands first focusing on the movement planning epoch (*Figure 6A*). As for M1, the LFP-latent dynamics correlations were

markedly frequency-dependent (*Figure 6B*), much stronger in the low and high frequencies, for which they greatly exceeded the surrogate control (two-sided Wilcoxon's rank sum test, p<0.001 for all comparisons). Interestingly, virtually all the datasets (10 of 12; pooled results in *Figure 6—figure supplement 1A*) exhibited a strong correlation at 12–25 Hz, which was notably absent in M1 despite the two regions having similar power spectra (*Figure 6—figure supplement 2A*), as well as power spectral density (*Figure 6—figure supplement 2B*) and variance (*Figure 6—figure supplement 2C*) in each LFP band.

Similar to M1, PMd is involved in both movement preparation and execution. Thus, we tested whether the LFP-latent dynamics correlation profile was preserved as in M1, and found it to be stable between movement planning and execution (R²=0.79; p<0.001, F-test; *Figure 6C*; *Figure 6—figure supplement 2D, E* show that LFP power remains constant across these two epochs too). During the inter-trial period when PMd neurons become less active, the correlations dropped, though less markedly than for M1 (*Figure 6D*). Therefore, the frequency-dependent relationship between LFPs and latent dynamics changes across cortical regions but, within a region, remains stable across different behaviour-related processes those regions are involved in.

Last, we studied whether the LFP's ability to predict the upcoming movement (*O'Leary and Hatsopoulos, 2006*) was also frequency-dependent. The performance of classifiers that predicted the upcoming target from the instructed delay activity (*Gallego et al., 2020*; *Santhanam et al., 2009*) (Methods) varied dramatically across LFP bands (*Figure 6E*; individual examples in *Figure 6—figure supplement 1B*): high-frequency bands were almost as accurate as the latent dynamics, while classifier accuracy at low frequencies was worse and quite variable across monkeys (~50% accuracy for Monkey C; close to chance for Monkey M). Interestingly, not all bands that were significantly correlated with the latent dynamics also predicted the upcoming target (*Figure 6—figure supplement 1C*); most notably, despite being among the most correlated bands, 12–25 Hz prediction accuracy was barely above chance.

## The LFP-latent dynamics correlation profiles are different for area 2 of primary somatosensory cortex

While LFPs in PMd and M1 primarily relate to preparing and generating movement, oscillations in area 2 of somatosensory cortex likely reflect somatosensory feedback processing (*Witham et al., 2007*;

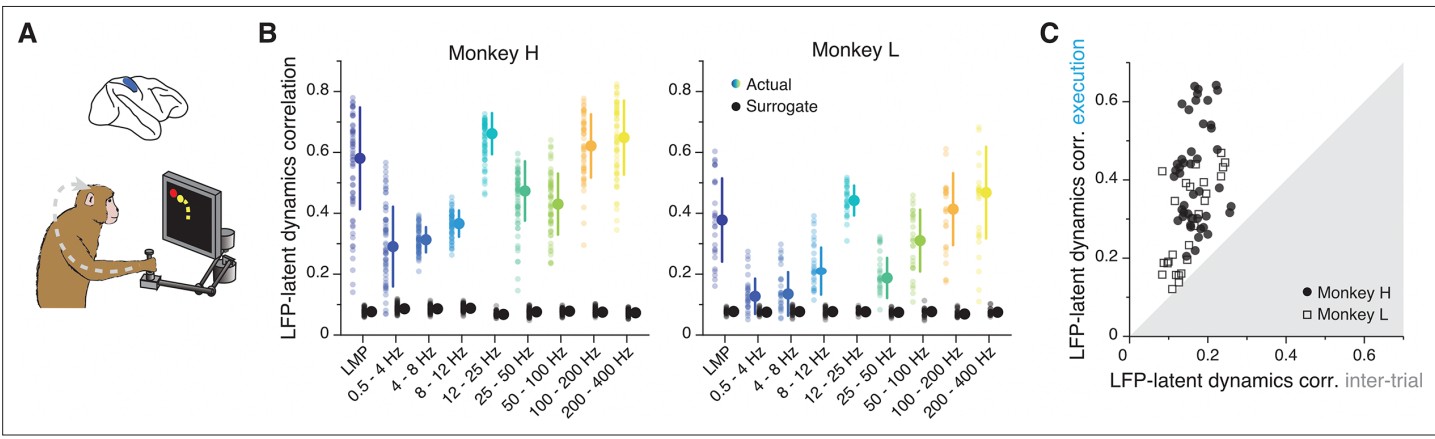

**Figure 7.** Similarity between area 2 latent dynamics and each local field potential (LFP) band during somatosensory feedback processing. (**A**) We focused on movement execution, when area 2 receives proprioceptive input about the state of the limb. (**B**) Correlation between each LFP band and the latent dynamics (coloured markers) during one representative session from each area 2 monkey; same format as *Figures 3B and 6B* (n=60, 28 for Monkeys H and L, respectively). Note the clear frequency-dependent relationship between LFP and latent dynamics. Error bars, median ± s.d. (**C**) Comparison between the LFP-latent dynamics correlations during movement execution and the inter-trial period; same format as *Figures 5C and 6D*. Note the marked decrease in LFP-latent dynamics correlation during the inter-trial period.

The online version of this article includes the following figure supplement(s) for figure 7:

**Figure supplement 1.** Similarity between area 2 latent dynamics and each local field potential (LFP) band, and decoding hand kinematics from area 2 LFPs.

**Figure supplement 2.** Region-specificity of the local field potential (LFP)-latent dynamics correlation profile.

*Baker et al., 2006*), perhaps in combination with efference copy signals (*Witham et al., 2010*). We thus asked whether the LFP-latent dynamics correlation profile in area 2 is different from that of motor regions (*Figure 7A*). Both the low- and high-frequency LFP bands of area 2 were strongly correlated with the latent dynamics (*Figure 7B*; two-sided Wilcoxon's rank sum test, p<0.001 for all comparisons; pooled results in *Figure 7—figure supplement 1A*), similar to the profile shown above in both M1 and PMd. Intriguingly, we observed a strong, significant correlation at 12–25 Hz (and for Monkey H also at 25–50 Hz), which we previously observed in PMd but not in M1. Again, all LFP-latent dynamics correlations dropped dramatically during the inter-trial period (*Figure 7C*), when monkeys did not move, and area 2 neurons go practically silent (*London and Miller, 2013*).

As in the other regions, we examined whether the ability to make predictions of the monkey's behaviour from each LFP band was correlated with the strength of its association with the latent dynamics. We again used linear decoders, in this case to predict the past state of the limb, consistent with the anticipated causality with area 2 (*Gallego et al., 2017*; *Chowdhury et al., 2020*) (Methods). As was the case for M1, the low- and high-frequency bands provided the best predictions (*Figure 7— figure supplement 1B, C*). Yet, contrary to M1 (though similar to PMd), the strength of the correlation between an LFP band and the latent dynamics did not predict how well that band could be used to decode movement (*Figure 7—figure supplement 1D*). Therefore, we have found robust region-specific LFP-latent dynamics correlation profiles that remain stable for different processes underlying behaviour (*Figures 3B, 6B and 7B*; *Figure 7—figure supplement 2*), and whose correspondence to behavioural decoding is strongest for M1.

## LFP-latent dynamics correlations are not predicted from single neuron features

Our results indicate stable, region-specific, and frequency-dependent relationships between LFP bands and latent dynamics. However, the observed correlations could simply have arisen due to trivial relationships between LFP and single neuron spiking activity (*Pesaran et al., 2002*; *Perel et al., 2015*; *Witham and Baker, 2012*; *Ray et al., 2008*; *Ray and Maunsell, 2011*; *Rule et al., 2017*; *Murthy and Fetz, 1996b*), rather than being dependent on population-wide latent dynamics. To directly address this, we asked whether LFP bands that were well correlated with the latent dynamics mostly captured the firing rates of the neurons recorded on that electrode. If that were the case, the correlations

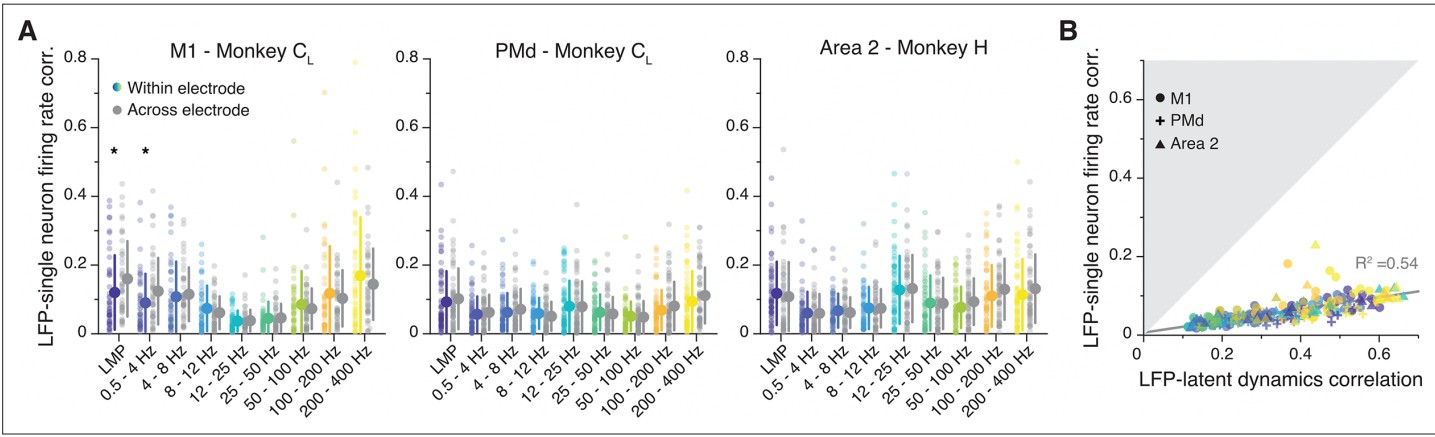

**Figure 8.** Single neuron features do not account for the local field potential (LFP)-latent dynamics correlations. (**A**) Comparison between the correlation of each LFP band with the firing rate of single neurons on the same electrode (coloured), and the activity of single neurons on a different electrode (grey). Each plot shows a representative session from one monkey implanted in each of the three studied regions. Error bars, median ± s.d. (n=50, 63, 60 for the primary motor cortex (M1), dorsal premotor cortex (PMd), and area 2 session, respectively). *p<0.001 two-sided Wilcoxon's rank sum test. (**B**) Comparison between the LFP-latent dynamics correlations and the correlation between the LFP and the firing rate of neurons on the same electrode. Data from all sessions and monkeys, shown using region-specific markers that indicate the session median (legend). Markers are colour-coded according to frequency as in A. Grey line, linear fit to pooled data and goodness of fit ($R^2$). Note how the LFP-latent dynamics correlations are much stronger than the correlations between LFPs and the activity of neurons on the same electrode.

The online version of this article includes the following figure supplement(s) for figure 8:

**Figure supplement 1.** Additional data: Single neuron features do not account for the local field potential (LFP)-latent dynamics correlations.

between LFPs and the firing rate of neurons on the same electrode ('within-electrode correlation') should exceed the analogous correlations between LFPs and the firing rates of neurons on different electrodes ('across-electrode correlation') (Methods). In fact, most within-electrode correlations for M1, PMd, and area 2 were very low, statistically no different from the across-electrode correlations (two-sided Wilcoxon's rank sum test, p≥0.1, except for M1 LMP (p=0.03) and 0.5–4 Hz (p=0.05); *Figure 8A*), suggesting that the observed similarities between LFPs and latent dynamics cannot be predicted from the single neuron activity.

As a second control, we compared the LFP-latent dynamics correlation directly to the correlation between the LFP and the single neuron activity on that electrode. Notably, the LFP-single neuron correlations were always much lower than the LFP-latent dynamics correlations (*Figure 8B*). Although there was a linear association between the two ($R^2$=0.54; p<0.001, F-test) the slope was very low ($\beta_1$=0.15), which indicates that even for high LFP-latent dynamics correlations, the LFP-single neuron correlations were still rather weak. The LFP-single neuron activity correlations remained low after we 'denoised' the single neuron firing rates by subtracting their projection outside the neural manifold, seeking to eliminate the components of the single neuron activity that are not shared across the population (Methods) (*Figure 8—figure supplement 1A–C*). Combining all the neurons on the same electrode as 'multi-units' also did not change the results (*Figure 8—figure supplement 1D,E*). These controls show that specific LFP bands capture population-wide features of the latent dynamics that contribute to their generation in a region-specific, stable manner.

## Discussion

LFPs are intriguing neural signals: changes in power within specific LFP frequency bands correlate well with behavioural and cognitive processes such as initiating a movement (*Sanes and Donoghue, 1993*; *Perel et al., 2015*; *Donoghue et al., 1998*; *Witham et al., 2007*; *O'Leary and Hatsopoulos, 2006*) or choosing among different stimulus-response rules (*Buschman et al., 2012*), yet their relationship with the activity of the neurons driving these processes has long been elusive. LFP biophysics are determined by circuit architecture (*Einevoll et al., 2013*) and the correlation among synaptic inputs to neural populations (*Lindén et al., 2011*). Since both of these also help define the neural covariance structure and shape the latent dynamics, we hypothesised that there should be a fundamental relationship between LFPs and latent dynamics. Here, we report, for each of three different regions of monkey sensorimotor cortex, that associations between LFPs and latent dynamics are stable and frequency-dependent (*Figures 3B, 6B and 7B*; *Figure 7—figure supplement 2*). They are also region-specific, as expected if main function and circuit biophysics play a role in determining them.

### Relation to previous single neuron studies

Previous studies investigated the relationship between single neuron activity and LFP power within different frequency bands. Spanning a broad range of regions, from primary visual to motor cortices, they present a puzzling story in which only the medium-high frequency LFPs (typically, 40–80 Hz) are associated with the activity of single neurons or multi-units (*Pesaran et al., 2002*; *Perel et al., 2015*; *Witham and Baker, 2012*; *Ray et al., 2008*; *Ray and Maunsell, 2011*; *Rule et al., 2017*; *Murthy and Fetz, 1996b*). The lack of single neuron-LFP correlations at low frequencies seems at odds with biophysical models predicting shared synaptic inputs – expected to drive single neuron activity – to be most strongly represented in the low-frequency LFPs (*Łęski et al., 2013*). Studies of M1 multielectrode LFP recordings align better with this prediction, since both the high-frequency and low-frequency LFPs allow muscle activity (*Flint et al., 2012*) and movement kinematics (*Bansal et al., 2012*; *Stavisky et al., 2015*) to be decoded accurately (*Figure 4*). Assuming that a large component of the firing of M1 neural populations relates to the motor commands, the ability to predict behaviour from the low-frequency LFPs hints at a relationship between the two. Such a relationship has been recently modelled as a 'mode' that captures behaviour-related dynamics that are shared between the low- and high-frequency motor cortical LFPs and the neural population activity (*Abbaspourazad et al., 2021*).

Here, we present a novel approach that explicitly quantifies the relationship between each LFP band and the population-wide latent dynamics. This allows us to show, directly, that low-frequency LFP bands are indeed correlated with the latent dynamics for each of primary motor (*Figure 3B*), dorsal premotor (*Figure 6B*), and somatosensory cortices (*Figure 7B*). We also extend our investigation

beyond movement execution by studying motor planning, to demonstrate that the LFP-latent dynamics correlation profiles remain stable throughout different processes of behaviour (*Figures 5A, B and 6C*). This result constitutes an important control. Since there is no overt movement during the planning epoch, the robust LFP-latent dynamics correlations during the instructed delay period indicate that their association is not only driven by large common inputs reflecting motor commands: it extends to more cognitive processes such as planning an action.

Reassuringly, the LFP-latent dynamic correlations drop during the inter-trial intervals, when monkeys are not engaged in the task and the synaptic currents across the neural population presumably become much weaker, as suggested by the decrease in single neuron firing rates (*Figure 5—figure supplement 1A, B*) and the disappearance of large population-wide covariation patterns (*Figure 5—figure supplement 1C*). This drop suggests that the LFP-latent dynamics correlations are a consequence of strong, coordinated synaptic currents across the circuit. Interestingly, such a decrease in correlation during the inter-trial period in PMd was not as marked as in M1, perhaps because PMd may retain some aspects of the task during this period, as recently shown for the prefrontal cortex (PFC) (*Mehta et al., 2020*; *Maggi and Humphries, 2022*). Indeed, PMd activity but not M1 activity reflects the probability of possible upcoming reach directions (*Glaser et al., 2018*), and it also has stronger reward-related signals than M1 (*Ramkumar et al., 2016*). Under this interpretation, we would expect 'higher' brain regions such as PFC to exhibit large LFP-latent dynamics correlations in the absence of overt behaviour, provided that the animal is deliberating or reminiscing.

## A neural population view is necessary

A critical aspect of our results is that the stable LFP-latent dynamics correlation profiles cannot be predicted based on the relationship between LFPs and single neurons. We interpret this based on the biophysics underlying LFP generation (*Einevoll et al., 2013*; *Buzsáki et al., 2012*; *Pesaran et al., 2018*). Biophysical models suggest that strong LFP signals may require correlated synaptic inputs to the neurons generating the electric fields, with the spatial extent of such correlated inputs determining the magnitude of the LFP (*Lindén et al., 2011*). These correlated inputs will lead to correlated neural firing patterns that will be captured in the latent dynamics computed with PCA, but much less so in the discharge of any given neuron. Thus, our finding of robust LFP-latent dynamics correlations is likely driven by the biophysical mechanisms of LFP generation and requires explicitly computing the dominant neural covariance patterns, since these seem to be largely determined by shared synaptic inputs across the population. In contrast, the activity of individual neurons is determined by a high-dimensional set of inputs affecting any given neuron through unknown weights, blurring their relationship with the LFP.

For a given level of correlated synaptic input, the LFP propagates in a frequency-dependent manner, with its spatial reach decreasing quite markedly at higher frequencies (*Łęski et al., 2013*). Yet, high-frequency components, especially broadband signals at frequencies >50 Hz, seem also to reflect local neural spiking (*Ray and Maunsell, 2011*). It is thus possible that the robust associations defining our roughly V-shaped LFP-latent dynamics correlation profiles (*Figures 3B, 6B and 7B*) have two different sources: at low frequencies, they may be dominated by the correlated synaptic inputs leading to the emergence of low-dimensional latent dynamics in the sensorimotor cortices during behaviour; at high frequencies, they may reflect a combination of correlated synaptic input across the population and more local neural spiking. In agreement with this, 'multi-unit' firing rates at 100–200 Hz and 200–400 Hz are more strongly correlated with the LFPs than single neuron firings rates (*Bansal et al., 2012*; *Perel et al., 2015*) (compare *Figure 8B* and *Figure 8—figure supplement 1D*), although still less so than the LFP-latent dynamics (*Figure 8—figure supplement 1E*).

## Function as well as circuit biophysics influences the differences between cortical areas

Together with functional specialisation, the importance of circuit biophysics in determining the LFP properties may also explain the differences in LFP-latent dynamics correlations across regions. Likely due to their varying evolutionary past (*Kaas, 2004*), each of the three cortical regions we studied has different neuron types, layer organisation, and local and long-range connectivity patterns (*Harris and Shepherd, 2015*), all of which influence the biophysics of LFP generation (*Einevoll et al., 2013*; *Lindén*

*et al., 2011*; *Buzsáki et al., 2012*; *Łęski et al., 2013*). Because of this, it is reasonable for the 'mapping' between latent dynamics and LFPs captured by our correlation profiles to be region-dependent.

The most striking difference in LFP-latent dynamics correlation profiles across regions appears at the mid-range 12–25 Hz band. Surprisingly, there is not a clear rostro-caudal gradient in their relationship; instead, the correlation is high in PMd (*Figure 6B*) and even higher in area 2 (*Figure 7B*), but extremely low in the intermediately located M1 (*Figure 3B*). What is common across all three regions is that the 12–25 Hz band is never a good predictor of movement parameters, even when the LFP-latent dynamics correlations reach 0.5–0.6 (*Figures 4C and 6E*; *Figure 7—figure supplement 1B,C*). This lack of correlation with movement parameters is consistent with studies of M1 LFPs during movement (*Sanes and Donoghue, 1993*; *Murthy and Fetz, 1996a*), which describe a marked decrease in LFP 12–25 Hz power prior to movement initiation (*Sanes and Donoghue, 1993*; *Perel et al., 2015*; *Donoghue et al., 1998*; *Witham et al., 2007*), with some transient oscillations only during specific tasks. These oscillations often appear when monkeys pick up treats, explore their surroundings, or perform relatively challenging finger tasks, but are very rare during more 'automatic' wrist tasks (*Sanes and Donoghue, 1993*; *Baker et al., 1997*; *Murthy and Fetz, 1996a*) – the latter being most similar to the centre-out reaching task we studied. Combined, these observations suggest that the 12–25 Hz LFP band may be a by-product of population activity related to attention (*Murthy and Fetz, 1996a*) or other high-level cognitive features such as keeping track of task structure. As such, 12–25 Hz LFP power could potentially be driven by the activity of neural populations in regions projecting to the sensorimotor cortices, rather than by the activity of local populations (*Herreras, 2016*). Indeed, LFP 12–25 Hz power in PFC modulates as monkeys select between stimulus-response rules (*Buschman et al., 2012*), suggesting that PMd 12–25 Hz oscillations could be correlated with task structure. Analogously, parietal cortical LFP 12–25 Hz power changes as monkeys choose between movement types based on visual cues (*Scherberger et al., 2005*), again suggesting a potential origin of area 2 12–25 Hz oscillations related to task structure.

## Conclusion

A wealth of studies addressing regions spanning the entire brain have reported on frequency-dependent changes in LFP power that correlate with features of behaviour. Yet, the relationship between the LFPs and the activity of neurons driving behaviour has remained elusive. Here, we show that, likely due to the biophysics of their generation, LFPs are fundamentally related to the shared patterns that dominate the activity of a neural population, rather than to the activity of the neurons themselves. While this frequency-dependent association varies across the sensorimotor cortex, for a given cortical region and frequency band it is stable throughout the behaviour.

We anticipate that our approach will uncover stable LFP-latent dynamics correlations not only within the sensorimotor cortices but beyond, as low-dimensional latent dynamics have been found across many cortical and subcortical brain regions (*Gallego et al., 2017*; *Keemink and Machens, 2019*). We further expect to uncover stable LFP-latent dynamic correlations during cognitive tasks, since the two are correlated during 'abstract' movement planning. Identifying these relationships offers an exciting opportunity to integrate studies based on either type of signals, including a large body of field recordings in neurological patients (*Kelley et al., 2018*; *Smith et al., 2019*; *Zheng et al., 2017*; *Vaz et al., 2019*; *Anumanchipalli et al., 2019*; *Miller et al., 2018*; *Wang et al., 2018a*; *Méndez-Bértolo et al., 2016*).

## Methods
### Behavioural task

Four monkeys (males, *Macaca mulatta*) were trained to sit in a primate chair and perform a centre-out reaching task using a planar manipulandum. All monkeys completed the task with the hand contralateral to the implanted hemisphere. During the task, the monkey started a trial by bringing the cursor to a target in the centre of the workspace. After a variable waiting period, the monkey was presented with one of eight outer targets (four for Monkey L), which were equally spaced along a circle of 6–8 cm radius. Monkeys C and M were trained to wait for the auditory go cue in the centre during a variable delay period of 0.5–1.5 s in which the target remained visible. Monkeys H and L were not subjected to this delay period. To receive a liquid reward, the monkey had to move the cursor into the outer target

within 1 s; Monkeys M and C were required to hold the cursor there for 0.5 s, whereas Monkeys H and L were only required to hold for 0.1 s, to ensure they had decelerated and ended the trial within the target. To start a new trial, the monkeys had to return the cursor to the central target. During the task, the endpoint position of the manipulandum was recorded at a sampling frequency of 1 kHz; the timing of the task events was digitally logged. Hand velocity was computed as the derivative of the hand position. We analysed the following number of sessions for each monkey: Monkey $C_L$: 6; Monkey M: 6; Monkey $C_R$: 4; Monkey H: 5; Monkey L: 3. In all the analyses, we only considered successful trials (an average of 307±221 trials per session; mean ± s.d.).

## Neural implants and recordings

All surgical and experimental procedures were approved by the Institutional Animal Care and Use Committee (IACUC) of Northwestern University under protocol #IS00000367. We recorded the spiking activity of neural populations and intracortical LFPs using chronically implanted 96-channel Utah electrode arrays in different regions of cortex. Monkey M was implanted in the M1 and PMd of the right hemisphere. Monkey C was implanted two times: first, he received an array in the right M1 (denoted by $C_R$ throughout the text) and then, in a second procedure, he received implants in the left M1 and PMd (denoted by $C_L$). Monkeys H and L were implanted in area 2 of primary somatosensory cortex of the left hemisphere.

Neural activity was acquired using a Cerebus system (Blackrock Microsystems, Salt Lake City, UT) at 30 kHz sampling frequency. The recordings on each channel were band-pass filtered (250–5000 Hz), and then converted to spike times based on threshold crossings. The threshold was selected according to the root mean squared (RMS) activity in each channel (Monkeys C and M: 5.5×RMS; Monkeys H and L: 5×RMS). The spike times were later sorted to identify putative neurons using specialised software (Offline Sorter v3, Plexon, Inc, Dallas, TX). The number of identified neurons varied across cortical area (M1: 60±20; PMd: 128±52; S1: 47±25; all mean ± s.d.). We simultaneously recorded the LFP on each channel at a sampling frequency of 1–2 kHz.

## Behavioural epochs

For each trial, we isolated different epochs in the task to identify different 'aspects' of the behaviour, namely movement preparation and execution. These windows were adjusted to the behavioural idiosyncrasies of each monkey; however, our results did not change qualitatively within reasonable changes to these windows. For Monkeys C and M, we used a movement execution window starting 120 ms before movement onset and ending 420 ms after onset. For Monkeys H and L, the movement execution window started at the go cue, and finished 600 ms after it. For the monkeys subjected to an instructed delay period (Monkeys C and M), we defined a movement preparation window starting 390 ms before movement onset and finishing 60 ms after movement onset. Finally, we also examined neural activity during the inter-trial period – when monkeys did not move the manipulandum – to study the relationship between neural population activity and LFP when monkeys are not engaged in a behaviour using a window starting 540 ms before target presentation and finishing at target presentation.

## LFP processing

The LFP signals were first de-referenced by computing their common average, and then centred at zero by subtracting their average over time for the entire session. Power line interference was removed using a notch filter (zero-phase Butterworth, second order, $f_c$ = 60 Hz). These pre-processed LFPs were then filtered (zero-phase Butterworth, third order) at the following frequency bands: 0.5–4 Hz, 4–8 Hz, 8–12 Hz, 12–25 Hz, 25–50 Hz, 50–100 Hz, 100–200 Hz, and 200–400 Hz. The band power of the LFP filtered at each frequency band was computed with an overlapping 50 ms window and 10 ms step. Since a 50 ms window does not cover the whole period of the frequency bands with minimum frequency less than 20 Hz, the band power at those frequency bands was computed using a window size of $1/f_{min}$ seconds, maintaining the 10 ms step. All LFP pre-processing procedures follow *Stavisky et al., 2015*. We also computed the so-called LMP by calculating the moving average of the LFP (de-referenced, detrended, and after removing the power line interference) using the same window setting as for the band power. Note that our results held when we varied the LFP frequency

bands within a reasonable range (*Figure 3—figure supplement 4B*; details in 'Additional analyses including controls' below).

Although we recorded LFPs from all 96 array channels in the arrays, we only considered in the analyses the LFPs from channels containing at least one putative neuron clearly identified through manual spike sorting, resulting in 46±16 (mean ±s.d.; ranging from 15 to 77) LFP channels per session. We subsampled the post-processed LFP signals to 30 ms bins, to match the bin size used for the single neuron activity, and smoothed them using a Gaussian kernel (s.d.: 50 ms) as we did for the single unit firing rates (see below).

## Neural population latent dynamics

To characterise the dynamics of the neural population activity, we first computed the smoothed firing rate for each putative single neuron by applying a Gaussian kernel (s.d.: 50 ms) to the binned square-root-transformed firings (30 ms bins). Although we excluded neurons with low firing rate (<1 Hz mean firing rate across all bins), we did not perform any additional preselection, for example, based on directional tuning. For each behavioural epoch (*Elsayed et al., 2016*) (execution or planning, when present) in each session, this produced a neural data matrix $\mathbf{X}$ of dimension $n$ by $T$, where $n$ is the number of neurons and $T$ is the total number of time points from all concatenated trials; thus, $T$ is the number of trials per session × the number of points per trial.

We represented the activity of the $n$ identified neurons during an epoch in a session in an $n$-dimensional neural state space, in which each axis represents the smoothed firing rate of one neuron. Within this space, we computed the low-dimensional manifold that spans the dominant neural covariation patterns (*Gallego et al., 2017*; *Cunningham and Yu, 2019*) by applying PCA on $\mathbf{X}$. We defined an $m$-dimensional neural manifold by keeping only the leading $m$ principal components. Based on previous work by our group and others (*Gallego et al., 2020*; *Churchland et al., 2010*; *Elsayed et al., 2016*), the number of components $m$ considered for each area were: $m$=10 for M1, $m$=15 for PMd, and $m$=8 for area 2; for those sessions with an instructed delay period, we used the same manifold dimensionality during the planning and execution epochs. Note that our results held across a wide range of dimensionality values $m$ (*Figure 3—figure supplement 3*). We computed the neural population latent dynamics by projecting the smoothed firing rates of the $n$ neurons onto the neural manifold. This produced a data matrix $\mathbf{L}$ of dimension $m$ by $T$, where $m$ is the dimensionality of the manifold.

## Alignment of latent dynamics and LFPs

We hypothesised that there should be an association between LFPs and latent dynamics, and that such an association would be region-specific, frequency-dependent, and stable during movement planning and execution. To address these hypotheses, we computed the similarity between the latent dynamics and the time-varying changes in LFP power in each band for each behavioural epoch, monkey, session, and cortical area. We used CCA, a method that quantifies the similarity between two sets of multidimensional signals by finding the linear combinations that, applied to each set, maximally correlates the resulting projections (*Bach and Jordan, 2002*); we refer to this process as 'aligning' an LFP band and the latent dynamics.

We started the alignment procedure by considering the concatenated latent dynamics, $\mathbf{L}$, and the concatenated LFP power in one electrode at one frequency band $b$, $\mathbf{F}_{b,i}$, where $b$ is one of the nine frequency bands listed above (including the LMP), and $i$ the electrode number; the vector $\mathbf{F}_{b,i}$ thus has dimension 1 by $T$. For each LFP band, we align separately the latent dynamics and the LFP power at one frequency band in each of the $k$ electrodes with at least one putative neuron, which yields $k$ canonical correlation coefficients, CCs, per LFP band. Throughout the paper, we summarise these distributions based on their mean ± s.d. (see, e.g., *Figures 3B, 6B and 7B*). For details on the implementation of CCA, see our previous work (*Gallego et al., 2020*).

To test whether the CCs between each LFP band and the latent dynamics capture a significant association between these two types of signals, we devised a control using TME, a method that generates surrogate neural population data preserving selected features of the original data (*Elsayed and Cunningham, 2017*). Here, we used TME to generate surrogate neural firing rates that preserved the covariance across neurons but that had different covariance across trials and over time (i.e., different dynamics) than the actual data. We then smoothed these surrogate firing rates with a Gaussian kernel

(s.d.: 50 ms), and applied PCA to obtain a matrix of surrogate latent dynamics, which were very similar to the actual latent dynamics based on their spectral content (*Figure 3—figure supplement 2*). To assess whether the experimentally measured CCs captured a significant association between an LFP band and the latent dynamics, we compared them to the CCs obtained after aligning the same LFP signals to the surrogate latent dynamics (shown in colour and in black, respectively, in *Figures 3B, 6B and 7B*).

## Decoding hand velocity from M1 and area 2 of somatosensory cortex movement activity

After quantifying to what extent each M1 and area 2 LFP band correlates with the latent dynamics, we asked how well it predicts behaviour. We computed Wiener filters (*Glaser et al., 2020*) to predict hand velocity from each of the nine LFP bands, including the LMP, as well as from latent dynamics. The filters included three bins of recent LFP history, for a total of 90 ms. These additional inputs seek to account for the intrinsic LFP dynamics and transmission delays and were only applied in this analysis. Given that M1 activity causes movement, for this region we included additional bins preceding the hand velocity signals. In contrast, since area 2 activity is largely driven by responses to the ongoing movement, for this region we included additional bins lagging the hand velocity signals. We further added a third-order polynomial at the output of the filter to compensate for nonlinearities (*Pohlmeyer et al., 2007*). Finally, to obtain accuracy values against which to compare the LFP decoders, we computed similar Wiener filters using the M1 or area 2 latent dynamics as inputs. Decoder prediction quality was measured based on the squared correlation coefficient ($R^2$) between the actual and predicted velocities. We built separate decoders for the X and Y hand velocities; as they had similar accuracy, we report their mean. The same approach was followed when predicting hand velocity from single channel LFPs (*Figure 4—figure supplement 1C,D*).

To avoid overfitting, we used 90% of the trials in the session to train the decoder and the remaining 10% to test it. We repeated this procedure 50 times, randomly selecting the non-overlapping training and testing trials in each iteration. We averaged the $R^2$ value across all the test blocks to obtain the final reported value.

## Predicting reach direction from PMd planning activity

We trained naïve Bayes classifiers (*Santhanam et al., 2009*) to predict the direction of the upcoming reach based on the pre-movement LFP activity in PMd. Similar to the velocity decoders, we built nine sets of LFP-based classifiers, each taking the LFP signals at one frequency band as inputs, as well as latent dynamics-based classifiers whose performance we used as reference. In all cases, we used PMd activity within a 450 ms window as input to the classifier. Within this window, we averaged all LFP activity (or latent dynamics) to obtain a single bin per trial, and then standardised it using a z-transform. Prediction accuracy was measured as the percentage of targets predicted correctly.

To protect against overfitting, we trained the classifier in 90% of the trials and tested the performance on the remaining 10%. As for velocity decoders, we repeated the procedure 50 times randomly selecting the non-overlapping training and testing trials in each iteration. We averaged the percentage of correct predictions to obtain the final reported value.

## Additional analyses including controls

We performed all the analyses in the paper using the standard LFP bands describe above. In addition, we also verified that our alignment results held even if the LFPs were filtered in different bands by repeating the analysis on partly overlapping bands that swept the entire LFP spectrum (*Figure 3—figure supplement 4B*). We chose pre-processing frequency bands that had width similar to that of the 'classic' LFP bands, which resulted in 53 bands with increasing width, ranging from 4 to 100 Hz. Filtering windows had a 50% overlap, and the following widths: 4 Hz between 0.5 and 14.5 Hz, 12 Hz between 10.5 and 26 Hz, 25 Hz between 12.5 and 57.5 Hz, 50 Hz between 25 and 120 Hz, and 100 Hz between 50 and 440 Hz.

To compute the LFP variance at each frequency band (*Figure 3—figure supplement 4C*, *Figure 6—figure supplement 2C*), we first normalised the post-processed LFP signals during the epoch of interest (preparation or execution) between 0 and 1 for each band and trial separately; this ensured a fair comparison of LFP variability across frequency bands. Then, we computed the variance

of these post-processed, normalised LFP signals, and calculated the LFP variance for each frequency band by pooling all the channels together.

We calculated the similarity between LFP bands by computing the pairwise correlation between the post-processed signals after concatenating all the trials from each session (*Figure 3—figure supplement 5A*). We followed the same approach to compare the frequency-dependent movement decoder predictions (*Figure 4—figure supplement 1B*).

In order to verify that the identified frequency-dependent associations between single LFP channel and neural population latent dynamics were not biased by applying PCA on the single neuron firing rates, we repeated the CCA on the 'latent LFP signals', which we identified by applying PCA on the post-processed multi-channel LFP recordings at each frequency band (*Figure 3—figure supplement 6*).

To study the LFP frequency content during different behavioural epochs, we computed the power spectral density of the post-processed LFP signals as the power of its direct Fourier transform divided by the frequency of each point (*Figure 6—figure supplement 2A*). To avoid any potential distortions due to power line interference (60 Hz) and its harmonics that could remain after pre-processing, we interpolated linearly the calculated power spectral density between –2 Hz and +2 Hz with respect to each harmonic. We followed the same approach when comparing the frequency content of M1 and PMd signals between movement preparation and execution (*Figure 6—figure supplement 2D*). To directly compare the power spectral density across regions (*Figure 6—figure supplement 2B*) or behavioural epochs (*Figure 6—figure supplement 2E*), we computed the mean of the power spectral density for each band.

To visually compare the LFP-latent dynamic associations across regions and behavioural epochs, we calculated for each session the mean canonical correlation per frequency band, and then averaged them per region (*Figure 5A* shows all three epochs for M1, and *Figure 7—figure supplement 2* summarises our results for all three sensorimotor cortical regions).

We devised two control analyses to test whether the similarities between LFP bands and latent dynamics could be explained by features of the single neuron activity. In the first control, we verified that the LFP on an electrode does not strongly reflect the activity of single neurons in that electrode. Thus, we verified that the relationship between LFP bands and single neurons on the same electrode did not exceed that between LFP bands and single neurons on different electrodes. We computed the Pearson correlation between the activity of each putative neuron and the power of each LFP band on that same electrode; this yielded nine distributions of $n$ correlation values. We then computed the Pearson correlation between each LFP band and the activity of a randomly selected neuron on a different electrode; this again yielded nine distributions of $n$ correlation values. Note that neurons detected on neighbouring electrodes were excluded to make sure the distance in the across-electrode group was not small. Direct comparisons between these paired sets of distributions allowed us to show that single neuron activity does not explain away our results (*Figure 8A*). As a second control, we compared directly, for each LFP band, the correlation between the LFP and the single neuron activity to the CC between the LFP and the latent dynamics (*Figure 8B*).

We repeated these controls after summing up the spikes of all putative single neurons on an electrode to form a 'multi-unit' (*Figure 8—figure supplement 1D,E*), and after 'denoising' the single neuron firing rates based on PCA (*Figure 8—figure supplement 1A–C*). This denoising consisted in subtracting from the smoothed firing rate of each neuron its projection outside the neural manifold, that is, the projections onto principal component (dimensions) $m+1$ to $n$. Notably, none of these manipulations changed the results qualitatively.

## Statistics

We applied statistical tests to compare results for the actual LFP-latent dynamics correlations and the surrogate LFP-latent dynamics correlations. In addition, we tested whether each of the LFP bands was more similar to the firing rates of each sorted single neuron or to the multi-unit activity on each electrode. For all these comparisons, we used a two-sided Wilcoxon's rank-sum test to compare the distributions, which were not necessarily assumed to be normal. When examining linear relationships between variables, model fits were compared to a null model in which only the intercept was different from zero using an F-test. Throughout all analyses we used a significance threshold of p<0.001. Sample sizes are reported in the corresponding figure captions; when a single number is reported, it refers to

all of the distributions plotted in that panel. No statistical methods were used to predetermine sample sizes, although our dataset included a large number of sessions and subjects, many more than typical studies in the field. The data analysis supporting our conclusions required no experimental intervention, so no randomisation into groups was needed and the experimenters were not blinded to the nature and goals of the experiment.

## Code availability

All analyses were implemented using custom Matlab (The Mathworks Inc) code. All code developed for this paper available on GitHub. Link: https://github.com/BeNeuroLab/Relationship_between_latent_dynamics_and_LFP; *Cecilia, 2022*.

## Acknowledgements

This work was supported in part by: grant F31-NS092356 from the National Institute of Neurological Disorder and Stroke, and grant T32-HD07418 from the National Institute of Child Health and Human Development (MGP); graduate research fellowship DGE-1324585 from the National Science Foundation, and grant T32-NS086749 from the National Institute of Neurological Disorder and Stroke (RHC); grants NS053603, NS074044, and NS095251 from the National Institute of Neurological Disorder and Stroke (LEM); and grant 2017-T2/TIC-5263 from the Community of Madrid, grant PGC2018-095846-A-I00 from the Spanish Ministry of Science and Innovation, grant EP/T020970/1 from the UKRI Engineering and Physical Sciences Research Council, and grant ERC-2020-StG-949660 from the European Research Council (JAG).

## Additional information

### Funding

| Funder | Grant reference number | Author |
| --- | --- | --- |
| National Institute of Neurological Disorders and Stroke | F31-NS092356 | Matthew G Perich |
| National Science Foundation | DGE-1324585 | Raeed H Chowdhury |
| National Institute of Neurological Disorders and Stroke | T32-NS086749 | Raeed H Chowdhury |
| National Institute of Neurological Disorders and Stroke | NS053603 | Lee E Miller |
| National Institute of Neurological Disorders and Stroke | NS074044 | Lee E Miller |
| National Institute of Neurological Disorders and Stroke | NS095251 | Lee E Miller |
| Comunidad de Madrid | 2017-T2/TIC-5263 | Juan Álvaro Gallego |
| Ministerio de Ciencia e Innovación | PGC2018-095846-A-I00 | Juan Álvaro Gallego |
| Engineering and Physical Sciences Research Council | EP/T020970/1 | Juan Álvaro Gallego |
| European Research Council | ERC-2020-StG-949660 | Juan Álvaro Gallego |

The funders had no role in study design, data collection and interpretation, or the decision to submit the work for publication.

## Author contributions
Cecilia Gallego-Carracedo, Software, Formal analysis, Validation, Investigation, Visualization, Methodology, Writing – original draft; Matthew G Perich, Resources, Data curation, Software, Investigation, Visualization, Methodology, Writing – original draft, Writing – review and editing; Raeed H Chowdhury, Resources, Data curation, Software, Investigation, Visualization, Writing – original draft, Writing – review and editing; Lee E Miller, Funding acquisition, Visualization, Writing – review and editing; Juan Álvaro Gallego, Conceptualization, Software, Supervision, Funding acquisition, Investigation, Visualization, Methodology, Writing – original draft, Writing – review and editing

## Author ORCIDs
Cecilia Gallego-Carracedo  http://orcid.org/0000-0002-0478-2359
Matthew G Perich  http://orcid.org/0000-0001-9800-2386
Raeed H Chowdhury  http://orcid.org/0000-0002-5934-919X
Lee E Miller  http://orcid.org/0000-0001-8675-7140
Juan Álvaro Gallego  http://orcid.org/0000-0003-2146-0703

## Ethics
All surgical and experimental procedures were approved by the Institutional Animal Care and Use Committee (IACUC) of Northwestern University under protocol #IS00000367.

## Decision letter and Author response
Decision letter https://doi.org/10.7554/eLife.73155.sa1
Author response https://doi.org/10.7554/eLife.73155.sa2

# Additional files

## Supplementary files
• Transparent reporting form

## Data availability
All data used for this paper are posted on Dryad (https://doi.org/10.5061/dryad.xd2547dkt). All analyses were implemented using custom Matlab (The Mathworks Inc) code. All code developed for this paper available on GitHub (https://github.com/BeNeuroLab/Relationship_between_latent_dynamics_and_LFP, copy archived at swh:1:rev:cbda8e2e6106f5eb5ff98e18a689c595179ac5db).

The following dataset was generated:

| Author(s) | Year | Dataset title | Dataset URL | Database and Identifier |
|---|---|---|---|---|
| Gallego-Carracedo C, Perich M, Chowdhury R, Miller L, Gallego J | 2022 | Local field potentials reflect cortical population dynamics in a region-specific and frequency-dependent manner | https://dx.doi.org/10.5061/dryad.xd2547dkt | Dryad Digital Repository, 10.5061/dryad.xd2547dkt |

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
