## [Editor Report]

This paper will be of interest to electrophysiologists, systems neuroscientists and neural engineers. The authors describe a framework for evaluating the comparison between LFP dynamics and spikes and perform this comparison for several datasets recorded from motor, premotor, and sensory areas of cortex in rhesus macaque monkeys. These results serve as an important benchmark for the information content of LFP recordings, which is relevant to data collection in neuroscientific investigations and to designing brain computer interfaces.

---

## [Decision Letter]

**Decision letter after peer review:**

Thank you for submitting your article "Local field potentials reflect cortical population dynamics in a region-specific and frequency-dependent manner" for consideration by *eLife*. Your article has been reviewed by 2 peer reviewers, and the evaluation has been overseen by Srdjan OSTOJIC as the Reviewing Editor and Ronald Calabrese as the Senior Editor. The reviewers have opted to remain anonymous at this stage.

Both reviewers were enthusiastic about the manuscript, but have suggested additional analyses and improvements to the text.

The Reviewing Editor has drafted a consolidated list of suggestions to help you prepare a revised submission.

Essential revisions:

Analyses:

1. Given the frequency-based analyses presented, more detailed characterization of the LFP spectra will greatly benefit the paper. A key question the authors should address is whether the frequency-dependence (and its variance across areas) is related to differences in power spectra across areas. They present an analysis suggesting their results are not simply explained by variance differences across bands, but there are no analyses to address power differences (and deviations from the 1/f "noise" spectrum).

2. It would be beneficial to include cross-frequency analyses (e.g. correlation/coherence between LMP and 200-400 hz). The discussion notes that there are potentially different sources of the LFP-latent correlations. More detailed comparisons of relationships within LFP frequency bands and with kinematics might begin to shed some light on this important issue and increase the manuscript's impact. This could be particularly interesting for PMd and area 2 where the 12-25hz band and higher frequencies are both correlated with latent activity, but only higher frequencies are predictive of behavior.

3. The "stable" claims focus on behavioral epochs while ignoring time. Are these relationships consistent across sessions?

4. A supplemental figure showing LFP-latent correlations vs. predictive accuracy on a single electrode basis would be beneficial.

5. The LFP-latent correlations appear larger in M1 than PMd. Is this statistically significant?

6. While the LFP-signal neuron firing correlations are clearly much smaller than LFP-latent correlations, Figure 8B and S8C/E clearly suggests a correlation between these two variables. This should be quantified and discussed.

7. The points below are focused around the central hang-up Reviewer 2 (R2) had while reading this paper: it seems like the motivation for this paper raises the question of what are the discrepancies in information content between LFP signals and latent dynamics estimated using spiking data. The authors do appropriately defend a narrow interpretation of their hypotheses, however, so the following comments should be seen as suggestions that the authors can choose to incorporate if they agree with the proposed rationale. If the authors choose to address these limitations, this would broaden the utility of these results for interpreting studies performed with LFP, or making concrete recommendations regarding future data collection. If the authors choose not to address this, please clarify the rationale and moderate the claims regarding the prescriptive relevance of the study for helping to contextualize LFP-only experiments.

– While R2 liked the questions and motivation for this paper, they were confused by the rationale central to details of the approach the authors used. It seems clear from the main text and methods that CCA is applied independently between the set of latent dynamics [#dim x #timepoints] and individual LFP channels [1 x #timepoints]. Initially, R2 assumed that the CCA was applied to multi-channel single-band LFP data [#channels x #timepoints], allowing for a comparison between two n-dimensional manifolds. This approach, instead, seems to look for the single best alignment between any one signal and one specific dimension of a (rotated) n-dimensional manifold. Using this as the primary metric gives us a sense for how well the signal for each single LFP band and channel is represented within the set of latent dynamics, but doesn't reveal what, if any, information is absent (or present) in LFP vis-a-vis latent dynamics, and vice versa. The correlation metric employed in this work does a good job of establishing that the analyzed LFP signals from individual frequency bands are well represented (or not) within the latent dynamical manifold. It does not, however tell us what information is lost by only considering LFP, which seems to be an important limitation of the approach.

– More specifically, it would be interesting to know if the combined signals from all LFP bands can be used to estimate the latent dynamics. Since the latent dynamics are assumed to be the ground truth for neural state, this seems like the most direct statement of a crucial question. If we form a matrix with dimensions [(#bands * # LFP channels) x #timepoints] and perform PCA on that matrix, does the resulting manifold look anything like the manifold estimated using PCA on spike data? If not, are the distortions the result of noise or a consistent bias?

– Regarding the point above, a complete investigation of the differences in information content between LFP and latent dynamics may be beyond the scope of this this paper, but methods such as probabilistic two-way partial least squares (PO2PLS, see [1]) might be appropriate for identifying components of shared variability, and private variability, between two matrices. In this case specifically, this approach (or another algorithm), could be used to identify components of the multidimensional data which are shared between LFP and the latent dynamics, and components which are only present in each modality's data alone. To be clear, it is up to the authors to choose whether they feel this suggestion is appropriate and within the scope of the work and aligned with their question.

– Regarding the points above, the presentation of Figure 3 is clear, but easy to misinterpret. By presenting 10 frequency bands (nine freq + LMP), compared with 10 latent dimensions, it gives the impression that the alignment procedure is being performed between the multi-band data and the latent dimensions. If this is indeed what was performed, then the text in the main manuscript and the Methods (page 17, Alignment of latent dynamics and LFPs) is confusing.

Text improvements:

8. Some further details of signal processing would be beneficial. For instance, the authors must be doing some form of normalization in power if the bands have similar variance across frequency, but it is not described in the methods. The computations used for the LFP variance analyses are also not explicitly defined, which is unfortunate given the importance of the control. There are several ambiguities in the methodological descriptions of LFP signal processing: 1) subtracting the average "over time" should have the time range defined (for each trial? Across a whole session?)), 2) "We subsampled the LFP signals…", should specify whether they mean broad-band LFP or (more likely) the post-processed signals (e.g., the estimated power time-series in each band).

9. Axes limits across figures need to be standardized. When showing data for two subjects side by side, axes limits should be the same. When comparing two variables to show relationships, the limits for each variable should be the same (i.e. square aspect ratio). These conventions are not systematically used across the manuscript (Example figures: Figures4D, 6B, 7B). Using a consistent scale across all plots of correlation etc. would be helpful for facilitating direct comparisons across areas.

10. The manuscript could be more precise in stating that the authors are looking at within-area relationships only.

11. The statement that β band activity (15-30 Hz) is "less informative" of behavior should be made more precise. Β band power is highly predictive of e.g. movement onset in well-trained tasks. The authors are presumably referring to predictive power of continuous-time kinematics.

12. The introduction's second hypothesis "Second, this relationship should be frequency-dependent, because only specific LFP bands are strongly correlated with the behavior." Somewhat conflicts with the presented data, consider revising this.

13. Figure 3's legend refers to "three reaches" but appears to show 4.

14. The authors state that this work "helps to bridge the wealth of studies reporting neural correlates of behavior using either type of recording," but stop short of making recommendations regarding future experiments or providing other prescriptive takeaways. While addressing this could easily expand beyond the scope of the work presented here, there are some questions that are natural to ask given the highlighted motivations. If little information is lost between LFP and latent dynamics, should we consider only recording LFP? For the design of neural sensors, recording LFP only would greatly reduce the bandwidth, and associated power and data storage or transmission requirements. Given a fixed power budget, one could use many more LFP channels than AP-band channels, suggesting a tradeoff between the number of LFP channels and the accuracy of the estimated neural state. It is perfectly fine for the authors to decide that this question is out of scope for this paper.

15. Figure 3B and 4B – the yellow lines for the 200-400 Hz band are hard to see. This plot uses the same color scheme employed in all other figures, and updating all figures is not a productive use of time. Altering the color for this one panel or thickening this line, or really anything else could help readers make a visual comparison to behavior and latent dynamics.

16. Figure S5 – Why is classification performance for reach direction so low for the low frequency bands when the correlation with latent dynamics is high, and the classification performance for latent dynamics is also high? Is this a concern or not?

17. Were time lags (e.g., ~50 ms) used when correlating behavior and LFP or latent dynamics?

18. Figure S5 – Missing panel label for panel D

References:

[1] Said el Bouhaddani, Hae-Won Uh, Geurt Jongbloed, Jeanine Houwing-Duistermaat. Statistical Integration of Heterogeneous Data with PO2PLS (https://arxiv.org/abs/2103.13490)

[Editors' note: further revisions were suggested prior to acceptance, as described below.]

Thank you for resubmitting your work entitled "Local field potentials reflect cortical population dynamics in a region-specific and frequency-dependent manner" for further consideration by *eLife*. Your revised article has been evaluated by Ronald Calabrese (Senior Editor) and a Reviewing Editor.

The manuscript has been improved but there are some remaining issues that need to be addressed, as outlined below. Specifically, we suggest tempering the claims that results reflect signatures of connectivity, as the evidence is indirect.

*Reviewer #1 (Recommendations for the authors):*

I thank the reviewers for their responses to my initial comments. Overall, the additional analyses have improved the manuscript. However, I have some lingering comments that I feel still need to be addressed before the paper would be suitable for publication.

1) One of the manuscript's primary claims is that LFP-latent correlations are "stable" within areas while being different between areas. These claims are the main basis of their interpretation that these relationships reflect biophysical properties of the cortical networks (e.g. cytoarchitecture). The claim of stable relationships focuses on comparing motor planning and execution task epochs. These task epochs appear to include partially overlapping time windows based on their methodological description, which seems like a potential confound that should be addressed. The time windows used are also different durations, which should be controlled for. Moreover, their results also show that LFP-latent relationships change (mostly disappearing) in inter-trial intervals. If these correlations truly reflect properties of circuit structure, I am unclear on why they would be task-dependent. This interpretational point needs significant clarification."

A related point is that the hypothesis stated in the introduction does not match well with the claims made by the results themselves. The introduction states "given that the synaptic connections remain stable in the short time scale spanning the preparation and execution of a movement, we expect the various LFP-latent dynamics associations should remain equally stable throughout these two processes underlying behaviour." This is a claim rooted in the underlying structure of neural circuits. In the section presenting the results, the authors say that seeing similar correlations in the movement and planning epoch: "indicates that the observed association is not a trivial epiphenomenal consequence of the frequency content of movement, but instead likely reflects underlying physiological processes related to the production of behaviour." which is not a claim about the underlying neural circuitry directly. The discussion then primarily focuses again on the goal of interpreting population activity as a reflection of neural circuity (e.g. connectivity).

The manuscript needs to be significantly strengthened if the authors wish to make claims in the introduction and discussion about their observations reflecting signatures of the underlying brain networks (cytoarchitecture and connectivity). Alternately, the introduction and discussion need to be revised to alter these claims.

---

## [Author Response]

Essential revisions:Analyses:1. Given the frequency-based analyses presented, more detailed characterization of the LFP spectra will greatly benefit the paper. A key question the authors should address is whether the frequency-dependence (and its variance across areas) is related to differences in power spectra across areas. They present an analysis suggesting their results are not simply explained by variance differences across bands, but there are no analyses to address power differences (and deviations from the 1/f "noise" spectrum).

We agree with the reviewers that these are all interesting analyses to add to the paper. Overall, the results, which we detail below, indicate that the reported differences in LFP-latent dynamics correlations across regions cannot be explained by large cross-regional differences in LFP spectral properties.

Figure 6—figure supplement 2A shows that the power spectra of the broadband LFPs were qualitatively similar across animals and sessions for any given region. This was also true across all three regions: a pairwise comparison of the band-specific power revealed strong similarities between regions (Figure 6—figure supplement 2B), with the only noticeable difference being a slight increase in power spectral density in the 12–25 Hz band for PMd and area 2 compared to M1.

All previous comparisons were performed during the movement execution epoch, since it was available for all three of the dorsal premotor, primary motor, and primary somatosensory cortices. We further verified that the broadband LFP spectrum did not change drastically between movement preparation and execution for both M1 and PMd (Figure 6—figure supplement 2D ). Figure 6—figure supplement 2E quantifies the similarity in LFP power at each frequency band across these two epochs, indicating a very strong similarity (except for a slight change in an outlier M1 session from Monkey C_L_).

To further investigate potential differences in LFP properties across regions, we computed the variance of each LFP band for PMd and area 2, to complement the M1 results included in Figure 3—figure supplement 4C of the original manuscript. This analysis, presented in Figure 6—figure supplement 2C, indicates that the LFP variance “profile” is very similar across each of the primary motor, dorsal premotor, and sensorimotor cortices. Moreover, it shows that across all three regions, the correlations with the latent dynamics cannot be simply explained by differences in variance across frequency bands: in fact, some of the LFP bands that are most strongly correlated with the latent dynamics, such as the LMP and the 200–400 Hz band, have relatively low variance compared to bands showing lower correlations.

2. It would be beneficial to include cross-frequency analyses (e.g. correlation/coherence between LMP and 200-400 hz). The discussion notes that there are potentially different sources of the LFP-latent correlations. More detailed comparisons of relationships within LFP frequency bands and with kinematics might begin to shed some light on this important issue and increase the manuscript's impact. This could be particularly interesting for PMd and area 2 where the 12-25hz band and higher frequencies are both correlated with latent activity, but only higher frequencies are predictive of behavior.

We have performed several analyses to address these interesting suggestions. First, we quantified the similarity between LFP bands by computing the correlation between all pairs of time-varying LFP signals from different frequency bands. Figure 3—figure supplement 5A and Author response image 1 summarises these results for each monkey. Overall, the only frequency bands that exhibited consistently large pairwise correlations between them (i.e., >0.4) were the 100–200 Hz and 200–400 Hz bands; all other correlations were much more modest. This is interesting because, in most cases, the LMP and the 200–400 Hz band were most strongly correlated with the latent dynamics, yet their pairwise correlations were in the 0.1–0.2 range.

**Author response image 1. sa2fig1:** **A**. Correlation between LFP bands for each of the M1 monkeys. Colour bar, correlation magnitude; upper triangle, mean correlation; bottom triangle, s.e.m. Note that the cross-band correlations are quite low, even for bands that are strongly correlated with the latent dynamics such as the LMP and 200–400 Hz. **B**. Same as A for Area 2.

Our previous analysis shows that, when compared directly, the time-varying power of two LFP bands with similar correlations with the latent dynamics may actually follow quite a different time course. To further probe the complexity of the LFP-latent dynamics correlations, we asked whether the strength of the correlations between LFP signals from two different bands recorded in the same electrode relate to their respective correlation with the latent dynamics. Figure 3—figure supplement 5B and Author response image 2 shows that this is not the case: many pairs of LFP bands with very different activity (|correlation| <0.2) showed quite similar canonical correlations (CCs) with the latent dynamics, with the difference in CC sometimes taking values ~0. Conversely, some pairs of strongly correlated LFP bands yielded quite different CCs with the latent dynamics, with differences as high as 0.3 (with 1 being the highest possible difference). Combined, these two analyses suggest that the similarities in strength of the LFP-latent dynamics correlations cannot be inferred from single channel LFP features alone.

After focusing on the complexity behind the frequency-dependent LFP-latent dynamics correlations, we investigated the similarity in movement-related information across frequency bands. We analysed whether the velocity predictions obtained from decoders trained on LFP signals from each frequency band resembled each other by measuring their correlation. Figure 4—figure supplement 1B and Author response image 3 shows that the movement predictions obtained from the bands that are the most predictive of movement kinematics (see Figure 4C and Figure 4—figure supplement 1A in the paper) are quite similar. This is perhaps not surprising since those bands (the low and high frequency bands) provide very good decoder performance, comparable to that of latent dynamics-based decoders. Conversely, the predictions obtained from LFP bands that were poor predictors of behaviour did differ among them, including for the 12-25 Hz band in PMd and area 2.

**Author response image 2. sa2fig2:** **A**. The correlation between LFP bands within the same electrode does not predict the correlation of each of these bands with the neural population latent dynamics. Legend, pair of bands being compared. Data for all sessions from each of the three M1 monkeys. **B**. Same as A for Area 2.

**Author response image 3. sa2fig3:** **A**. Correlation between hand velocity predictions obtained from decoders trained on each LFP band. Colour bar, correlation magnitude; upper triangle, mean correlation; bottom triangle, s.e.m. **B**. Same as A for Area 2.

3. The "stable" claims focus on behavioral epochs while ignoring time. Are these relationships consistent across sessions?

This is indeed an important point that we had not addressed in the manuscript. Figure 3—figure supplement 1B and Author response image 4 shows that the frequency-dependent relationship between LFPs and latent dynamics are stable across different sessions from the same monkey. There are a couple small “jumps” (for Monkey C_R_ and Monkey H) that we believe may have occurred due to potential electrode shifts between sessions, as the jump is most prominent when comparing two sessions from Monkey C_R_ that were 93 days apart. We also note that the jumps appear to be changes in gain, but do not change the broader pattern across frequency bands.

**Author response image 4. sa2fig4:** Stability of the LFP-latent dynamics correlation across different sessions from each monkey. Coloured squares represent the mean value across all LFP channels for each frequency band. Note that the frequency dependent correlations are very similar across sessions; the only noticeable change is a shift in magnitude between days seven and 100 for Monkey CR. Same as for Area 2.

4. A supplemental figure showing LFP-latent correlations vs. predictive accuracy on a single electrode basis would be beneficial.

On average, decoding hand kinematics from single LFP channels yielded considerably worse predictions than predicting from all the channels. However, the bands that were most predictive of behaviour such as the LMP or the higher frequency bands did have a subset of channels that yielded good decoding performance, although decoder accuracy was largely variable across electrodes (Figure 4—figure supplement 1C and D).

5. The LFP-latent correlations appear larger in M1 than PMd. Is this statistically significant?

The M1 LFP-latent dynamics correlations were indeed slightly greater than the PMd correlations, and these differences were statistically significant except for the 0.5–4 and 8–12 Hz bands (*P*<0.001; two-sided Wilcoxon’s rank sum test; see Author response image 5). However, these effects were always very small (the difference in means across regions never exceeded 0.1).

**Author response image 5. sa2fig5:** Comparison between the M1 and PMd frequency-dependent LFP-latent dynamics correlations. The plot shows the correlation between each LFP band and the latent dynamics; data pooled across all sessions and monkeys for each region. Violin: probability density for one frequency band; left distribution shows M1 and right distribution shows PMd; horizonal line, median. * *P*<0.001 two-sided Wilcoxon’s rank sum test.

6. While the LFP-signal neuron firing correlations are clearly much smaller than LFP-latent correlations, Figure 8B and S8C/E clearly suggests a correlation between these two variables. This should be quantified and discussed.

We have added linear fits to all the figures and discussed them in the text (Page 9).

7. The points below are focused around the central hang-up Reviewer 2 (R2) had while reading this paper: it seems like the motivation for this paper raises the question of what are the discrepancies in information content between LFP signals and latent dynamics estimated using spiking data. The authors do appropriately defend a narrow interpretation of their hypotheses, however, so the following comments should be seen as suggestions that the authors can choose to incorporate if they agree with the proposed rationale. If the authors choose to address these limitations, this would broaden the utility of these results for interpreting studies performed with LFP, or making concrete recommendations regarding future data collection. If the authors choose not to address this, please clarify the rationale and moderate the claims regarding the prescriptive relevance of the study for helping to contextualize LFP-only experiments.– While R2 liked the questions and motivation for this paper, they were confused by the rationale central to details of the approach the authors used. It seems clear from the main text and methods that CCA is applied independently between the set of latent dynamics [#dim x #timepoints] and individual LFP channels [1 x #timepoints]. Initially, R2 assumed that the CCA was applied to multi-channel single-band LFP data [#channels x #timepoints], allowing for a comparison between two n-dimensional manifolds. This approach, instead, seems to look for the single best alignment between any one signal and one specific dimension of a (rotated) n-dimensional manifold. Using this as the primary metric gives us a sense for how well the signal for each single LFP band and channel is represented within the set of latent dynamics, but doesn't reveal what, if any, information is absent (or present) in LFP vis-a-vis latent dynamics, and vice versa. The correlation metric employed in this work does a good job of establishing that the analyzed LFP signals from individual frequency bands are well represented (or not) within the latent dynamical manifold. It does not, however tell us what information is lost by only considering LFP, which seems to be an important limitation of the approach.

We agree with the reviewer that this is an interesting analysis, one that we had in fact performed prior to submitting the manuscript. We decided not to include it because the results are qualitatively similar to those obtained when examining the relationship between individual LFP signals and latent dynamics, which is the central question we are interested in.

As shown in Figure 3—figure supplement 6A, applying PCA to the multichannel LFP recordings revealed that, for all frequency bands as well as the LMP, a ten-dimensional “LFP manifold” explains ~85–99 % of the total LFP variance. We performed CCA between the ten-dimensional latent dynamics and the ten-dimensional “latent LFP signals” for each frequency band to quantify their frequency-dependent association. Figure 3—figure supplement 6B shows that these correlation profiles greatly resemble the correlation profiles between single channel LFPs and the latent dynamics (compare with Figure 3—figure supplement 1A in the paper). The only difference is that the values in Figure 3—figure supplement 6B are a bit greater, as it expected from this kind of PCA-based “denoising” of the multichannel LFPs.

After having established that the main trend of the frequency-dependent LFP-latent dynamics association is maintained regardless of whether one uses single LFP channel or latent LFP signals in the comparisons, we investigated whether each band’s relative amount of movement-related information also remains preserved. We built the same type of standard Wiener decoders we used in the manuscript to predict hand velocity. As for the decoders based on multi-input single LFP channels (see original manuscript), prediction accuracy exhibited a “V” shape, with the LMP and the low and high frequency bands providing very accurate predictions (Author response image 6).

**Author response image 6. sa2fig6:** Decoding movement kinematics from M1 “latent LFP signals”. **A**. Accuracy of each LFP band (colours) during an example representative session from Monkey C_L_. Markers: individual folds during cross-validation; error bars, median ± s.d. **B**. Same as A, with data pooled across all M1 monkeys and sessions. Violin, probability density for one frequency band; horizontal bars, median.

– More specifically, it would be interesting to know if the combined signals from all LFP bands can be used to estimate the latent dynamics. Since the latent dynamics are assumed to be the ground truth for neural state, this seems like the most direct statement of a crucial question. If we form a matrix with dimensions [(#bands * # LFP channels) x #timepoints] and perform PCA on that matrix, does the resulting manifold look anything like the manifold estimated using PCA on spike data? If not, are the distortions the result of noise or a consistent bias?

Applying PCA on this timepoints x [bands * channels] matrix reveals that the LFP variance is largely dominated by a few covariance patterns; for the same example M1 sessions used throughout the paper, the first ten principal components (PCs) explain >90 % of the total variance (Author response image 7). This value is slightly lower than the amount of variance explained by the first ten PCs for each of the LFP bands in isolation (Figure 3—figure supplement 6A), further indicating LFP bands are indeed different from each other (e.g., Figure 3—figure supplement 5A and Author response image 1).

**Author response image 7. sa2fig7:** **A**. Cumulative cross-band LFP variance explained as function of the numbers of dimensions considered. Note that as few as ten components explain most of the variance. Each plot shows the same example session from each M1 monkey shown throughout the paper. **B**. Canonical correlation between the ten-dimensional neural population latent dynamics and the “latent LFP bands * channels signals” as function of the number of the cross-band LFP dimensions considered. Mean (trace) ± s.d. (shaded surface) across the first four CCs for each crossband LFP manifold dimension.

When examining the relationship between these “latent LFP bands * channels signals” and the neural population latent dynamics, the canonical correlations (CCs) did not saturate even when most of the cross-band LFP variance was accounted for (e.g., when ten dimensions were considered). This was clearest for Monkeys M and C_R_: although the first ten dimensions explained almost all the cross-band LFP variance, their correlation with the latent dynamics did not exceed 0.2–0.3 (Author response image 7). Interestingly, the correlations with the latent dynamics were larger for those sessions in which the amount of cross-band LFP variance explained grew more slowly (example session from Monkey C_L_). This observation is in good agreement with previous analyses showing that large LFP signals in terms of their variance are often not well correlated with the neural population latent dynamics (e.g., Figure 6—figure supplement 2C). Finally, the correlations between the “latent LFP bands * channels signals” and the neural population latent dynamics plateaued at ~0.8, again suggesting that the latent dynamics do have additional “information” that is absent in the LFP.

– Regarding the point above, a complete investigation of the differences in information content between LFP and latent dynamics may be beyond the scope of this this paper, but methods such as probabilistic two-way partial least squares (PO2PLS, see [1]) might be appropriate for identifying components of shared variability, and private variability, between two matrices. In this case specifically, this approach (or another algorithm), could be used to identify components of the multidimensional data which are shared between LFP and the latent dynamics, and components which are only present in each modality's data alone. To be clear, it is up to the authors to choose whether they feel this suggestion is appropriate and within the scope of the work and aligned with their question.

We thank the reviewer for the suggestion. Methods like PO2PLS are certainly interesting and we would like to explore them in future studies, but we think that adding it would complicate our already quite long manuscript.

– Regarding the points above, the presentation of Figure 3 is clear, but easy to misinterpret. By presenting 10 frequency bands (nine freq + LMP), compared with 10 latent dimensions, it gives the impression that the alignment procedure is being performed between the multi-band data and the latent dimensions. If this is indeed what was performed, then the text in the main manuscript and the Methods (page 17, Alignment of latent dynamics and LFPs) is confusing.

The reviewer is correct in that the alignment is done separately for each LFP band. To avoid confusion, we have modified the schematic below “Align dynamics” indicating that we “Align LFP band to latent dynamics” by performing CCA between the “latent dynamics” and each “individual channel for each LFP band”**.**

Text improvements:8. Some further details of signal processing would be beneficial. For instance, the authors must be doing some form of normalization in power if the bands have similar variance across frequency, but it is not described in the methods. The computations used for the LFP variance analyses are also not explicitly defined, which is unfortunate given the importance of the control. There are several ambiguities in the methodological descriptions of LFP signal processing: 1) subtracting the average "over time" should have the time range defined (for each trial? Across a whole session?)), 2) "We subsampled the LFP signals…", should specify whether they mean broad-band LFP or (more likely) the post-processed signals (e.g., the estimated power time-series in each band).

Thanks for pointing this out. We have clarified all the issues raised by the reviewers and added a detailed description of all the new analyses.

9. Axes limits across figures need to be standardized. When showing data for two subjects side by side, axes limits should be the same. When comparing two variables to show relationships, the limits for each variable should be the same (i.e. square aspect ratio). These conventions are not systematically used across the manuscript (Example figures: Figures4D, 6B, 7B). Using a consistent scale across all plots of correlation etc. would be helpful for facilitating direct comparisons across areas.

We have both fixed the axis limits when showing the same variable for two subjects side by side (Figures 5A, 6B and 7B), and harmonised the axis limits for the same quantity across figures. However, we consider that when comparing two different variables in the same plot (i.e. Canonical Correlation Coefficients vs R-squared of decoder predictions), it is best if we keep the optimal axis limits for each variable individually, for a clearer visualization (Figure 4D).

10. The manuscript could be more precise in stating that the authors are looking at within-area relationships only.

We now mention explicitly throughout the paper that we focus on within area relationships.

11. The statement that β band activity (15-30 Hz) is "less informative" of behavior should be made more precise. Β band power is highly predictive of e.g. movement onset in well-trained tasks. The authors are presumably referring to predictive power of continuous-time kinematics.

Yes, that is what we meant. We have corrected this statement.

12. The introduction's second hypothesis "Second, this relationship should be frequency-dependent, because only specific LFP bands are strongly correlated with the behavior." Somewhat conflicts with the presented data, consider revising this.

Thanks for pointing this out. We have modified the text to clarify this statement.

13. Figure 3's legend refers to "three reaches" but appears to show 4.

Corrected.

14. The authors state that this work "helps to bridge the wealth of studies reporting neural correlates of behavior using either type of recording," but stop short of making recommendations regarding future experiments or providing other prescriptive takeaways. While addressing this could easily expand beyond the scope of the work presented here, there are some questions that are natural to ask given the highlighted motivations. If little information is lost between LFP and latent dynamics, should we consider only recording LFP? For the design of neural sensors, recording LFP only would greatly reduce the bandwidth, and associated power and data storage or transmission requirements. Given a fixed power budget, one could use many more LFP channels than AP-band channels, suggesting a tradeoff between the number of LFP channels and the accuracy of the estimated neural state. It is perfectly fine for the authors to decide that this question is out of scope for this paper.

We agree with the reviewer that these are important points to consider for translational studies. Indeed, LFP-based decoders using specific bands do seem to be almost as accurate as firing rate-based decoders (Figure 4C), in agreement with previous offline (Ref. 36–38 in the paper) and online studies (Ref. 39 in the paper). Moreover, although LFPs seem to be slightly worse than neural firing rates when predicting an intended movement during preparation (Figure 6E), this difference may become smaller during online control. Combined with the reduced power consumption and longer time span of LFP recordings, these results indeed indicate that intracortical LFP signals or even µECoG recordings (e.g., Chiang et al. Science Trans Med 2021) may be the optimal solution for BCI-based cursor control or BCI-based spellers. Although we find these issues interesting, we feel that these translational observations do not fit well within this current manuscript, and we would rather keep them for future work.

15. Figure 3B and 4B – the yellow lines for the 200-400 Hz band are hard to see. This plot uses the same color scheme employed in all other figures, and updating all figures is not a productive use of time. Altering the color for this one panel or thickening this line, or really anything else could help readers make a visual comparison to behavior and latent dynamics.

Thank you for pointing this out. As the reviewer says, changing the colour throughout the paper would take a substantial amount of work. We will keep this concern in mind for future manuscripts using the same colour palette.

16. Figure S5 – Why is classification performance for reach direction so low for the low frequency bands when the correlation with latent dynamics is high, and the classification performance for latent dynamics is also high? Is this a concern or not?

We think that this is an interesting point that further suggests that the large canonical correlations (CC) between LFPs and latent dynamics are not just a mere by-product of the behaviour. Indeed, for PMd, both the LMP and high frequency bands (100-200 Hz, and 200-400 Hz) have relatively large correlations with the latent dynamics, yet only the high frequency LFP bands are good predictors of the intended target. Thus, some of the correlation between, e.g., the LMP and the latent dynamics may be driven by signals not related to reach direction. The new analyses suggested by the Reviewers as part of Comment 2, further suggest that the underpinnings of the estimated LFPlatent dynamics correlations are complex.

Likewise, the 12–25Hz (β) band in area 2 of somatosensory cortex is highly correlated with the latent dynamics despite of it providing very low decoding performance compared to the latent dynamics. Thus, the similarity between these two types of signals arises not from them “encoding" continuous reach kinematics. Instead, they may reflect other aspects of behaviour, including signals in the neural population activity that are not modulated by the details of the movement (e.g., Kaufman et al., *eNeuro* 2016). Note that this figure has now become Figure 6—figure supplement 1.

17. Were time lags (e.g., ~50 ms) used when correlating behavior and LFP or latent dynamics?

Our M1 decoders that predict behaviour from the latent dynamics or the LFP included neural activity at time *t* plus past activity up to *t* – 90 ms (see Methods; values are taken from Morrow and Miller, *J Neurophysiol* 2003), with *t* being the current time.

18. Figure S5 – Missing panel label for panel D

Corrected (note that this figure has now become Figure 6—figure supplement 1).

References:[1] Said el Bouhaddani, Hae-Won Uh, Geurt Jongbloed, Jeanine Houwing-Duistermaat. Statistical Integration of Heterogeneous Data with PO2PLS (https://arxiv.org/abs/2103.13490)

[Editors' note: further revisions were suggested prior to acceptance, as described below.]

The manuscript has been improved but there are some remaining issues that need to be addressed, as outlined below. Specifically, we suggest tempering the claims that results reflect signatures of connectivity, as the evidence is indirect.

We sincerely thank the Editor and the Reviewers for their time, and we are very glad that they think the manuscript has been improved. In the previous versions of the paper, the claims about the impact of circuit connectivity on the relationship between local field potentials (LFP) and neural population latent dynamics were part of one of our original hypotheses (paragraphs three and four in the Introduction), and we had speculated in the Discussion about their influence in our results. We have now edited both of these sections as well as a part of the Results to temper our claims, in particular by also acknowledging the contribution of other circuit biophysical properties to LFP generation. In addition, in the Discussion, we have expanded on the interpretation of cross-regional differences based on their different function.

Reviewer #1 (Recommendations for the authors):I thank the reviewers for their responses to my initial comments. Overall, the additional analyses have improved the manuscript. However, I have some lingering comments that I feel still need to be addressed before the paper would be suitable for publication.1) One of the manuscript's primary claims is that LFP-latent correlations are "stable" within areas while being different between areas. These claims are the main basis of their interpretation that these relationships reflect biophysical properties of the cortical networks (e.g. cytoarchitecture). The claim of stable relationships focuses on comparing motor planning and execution task epochs. These task epochs appear to include partially overlapping time windows based on their methodological description, which seems like a potential confound that should be addressed. The time windows used are also different durations, which should be controlled for.

We are glad to see the Reviewer’s positive reaction to our changes.

The reviewer is right in that the movement preparation window and the movement execution window are partly overlapping—the final 40 % of the preparation window overlaps with the first 33 % of the movement onset window. We chose those windows to be consistent with our own previous work (e.g., Gallego, Perich et al. *Nature Neurosci* 2020), as well as with studies by others (e.g., Churchland et al. *J Neurosci* 2006, Sussillo et al. *Nature Neurosci* 2015). Critically, varying the preparation and execution windows within a reasonable range does not impact our observation that the relationship between LFPs and latent dynamics is stable within a region. Author response image 8 replicates our current summary figure (Figure 7–figure supplement 2) using non-overlapping movement preparation, movement execution, and inter-trial windows with equal lengths (-450 ms before movement onset to movement onset; movement onset to +450 ms after movement onset, and -450 ms before target presentation to target presentation, respectively; for area 2, we used a window +150 ms after movement onset to +600 ms after movement onset to account for that area primarily processing feedback about the state of the limb). These results are nearly indistinguishable from those included in the paper.

**Author response image 8. sa2fig8:** Region-specificity of the LFP-latent dynamics correlation profile when using non-overlapping “behavioural windows” of equal length. Line and shaded areas, mean ± s.e.m. across all session medians from all monkeys for each area (n = 16, 16, 8 for PMd, M1, and area 2, respectively). Note the striking similarity with Figure 7–supplement 2 in the paper.

Moreover, their results also show that LFP-latent relationships change (mostly disappearing) in inter-trial intervals. If these correlations truly reflect properties of circuit structure, I am unclear on why they would be task-dependent. This interpretational point needs significant clarification."

Regarding the LFP-latent dynamics correlations disappearing during inter-trial period, we agree that our interpretation was not completely clear. We have now edited the subsection *“The LFP-latent dynamics correlation profiles are stable between movement planning and execution”* to avoid any misunderstandings, clarifying that we expected the LFP-latent dynamic correlations to remain stable as long as the circuits biophysics remain stable and the region is “functionally engaged,” which for the motor cortices means generating behaviourally relevant activity. We trust that these edits and our rewriting of the Introduction and the Discussion as also requested by the Reviewer will make our interpretation of these results clearer.

A related point is that the hypothesis stated in the introduction does not match well with the claims made by the results themselves. The introduction states "given that the synaptic connections remain stable in the short time scale spanning the preparation and execution of a movement, we expect the various LFP-latent dynamics associations should remain equally stable throughout these two processes underlying behaviour." This is a claim rooted in the underlying structure of neural circuits. In the section presenting the results, the authors say that seeing similar correlations in the movement and planning epoch: "indicates that the observed association is not a trivial epiphenomenal consequence of the frequency content of movement, but instead likely reflects underlying physiological processes related to the production of behaviour." which is not a claim about the underlying neural circuitry directly. The discussion then primarily focuses again on the goal of interpreting population activity as a reflection of neural circuity (e.g. connectivity).The manuscript needs to be significantly strengthened if the authors wish to make claims in the introduction and discussion about their observations reflecting signatures of the underlying brain networks (cytoarchitecture and connectivity). Alternately, the introduction and discussion need to be revised to alter these claims.

We have tempered our claims related to the contribution of circuit connectivity since they are indeed not directly addressed by our data, and expanded on the potential contribution of a region’s main function as well as the overall biophysical properties of the circuit to interpret our results.